# Animacy semantic network supports causal inferences about illness

**Miriam Hauptman\*, Marina Bedny**

Department of Psychological & Brain Sciences, Johns Hopkins University, Baltimore, United States

**\*For correspondence:**
mhauptm1@jhu.edu

**Competing interest:** The authors declare that no competing interests exist.

## eLife Assessment

This study investigates the neural basis of causal inference of illness, suggesting that it relies on semantic networks specific to living things in the absence of a generalized representation of causal inference across domains. The main hypothesis is **compelling**, and is supported by **solid** methods and data analysis. Overall, the findings make a **valuable** contribution to understanding the role of domain-specific semantic networks, particularly the precuneus, in implicit causal inference about illness.

**Abstract** Inferring the causes of illness is a culturally universal example of causal thinking. We tested the hypothesis that making causal inferences about biological processes (e.g. illness) depends on the animacy semantic network. Participants (*n*=20) undergoing fMRI read two-sentence vignettes that elicited implicit causal inferences across sentences, either about the emergence of illness or about the mechanical breakdown of inanimate objects, in addition to noncausal control vignettes. All vignettes were about people and were linguistically matched. The same participants performed localizer tasks: language, logical reasoning, and mentalizing. Inferring illness causes, relative to all control conditions, selectively engaged a portion of the precuneus (PC) previously implicated in the semantic representation of animates (e.g. people, animals). Neural responses to causal inferences about illness were adjacent to but distinct from responses to mental state inferences, suggesting a neural mind/body distinction. We failed to find evidence for domain-general responses to causal inference. Causal inference is supported by content-specific semantic networks that encode causal knowledge.

## Introduction

A distinguishing feature of human cognition is our ability to reason about complex cause-effect relationships, particularly when causes are not directly perceptible (*Tooby and DeVore, 1987*; *Lagnado et al., 2007*; *Rottman et al., 2011*; *Muentener and Schulz, 2014*; *Sloman and Lagnado, 2015*; *Goddu and Gopnik, 2024*). When reading something like, *Hugh sat by sneezing passengers on the subway. Now he has a case of COVID*, we naturally infer a causal relationship between crowded spaces and the invisible transmission of infectious disease. Here we investigate the neurocognitive mechanisms that support such automatic inferences by studying causal inferences about illness.

Adults have rich, culturally specific causal knowledge about the invisible forces that bring about illness, from pathogen transmission to divine retribution (*Notaro et al., 2001*; *Raman and Winer, 2004*; *Lynch and Medin, 2006*; *Legare and Gelman, 2008*; *Legare et al., 2012*; *Legare and Shtulman, 2017*). In many societies, designated 'healers' become experts in diagnosing and treating disease (*Foster, 1976*; *Ackerknecht, 1982*; *Norman et al., 2009*; *Lightner et al., 2021*). Nonexpert adults routinely infer the causes of illness in themselves and others (e.g. *how did my friend get COVID?*).

Even young children think about illness in systematic ways, reflecting their burgeoning commonsense understanding of the biological world (*Wellman and Gelman, 1992*; *Keil, 1992*; *Inagaki and Hatano, 2006*). Young children attribute illness to contaminated food, contact with a sick person, and parental inheritance (*Springer and Ruckel, 1992*; *Kalish, 1996*; *Kalish, 1997*; *Keil et al., 1999*; *Notaro et al., 2001*; *Raman and Winer, 2004*; *Raman and Gelman, 2005*; *Legare and Gelman, 2008*; *Legare et al., 2009*; *DeJesus et al., 2021*).

Illness affects living things (e.g. people and animals) rather than inanimate objects (e.g. rocks, machines, houses). Thinking about living things (animates) as opposed to nonliving things (inanimate objects/places) recruits partially distinct neural systems (e.g. *Warrington and Shallice, 1984*; *Hillis and Caramazza, 1991*; *Caramazza and Shelton, 1998*; *Farah and Rabinowitz, 2003*). The precuneus (PC) is part of the 'animacy semantic network' and responds preferentially to living things (i.e. people and animals), whether presented as images or words (*Devlin et al., 2002*; *Fairhall and Caramazza, 2013a*; *Fairhall et al., 2014*; *Peer et al., 2015*; *Wang et al., 2016*; *Silson et al., 2019*; *Rabini et al., 2021*; *Deen and Freiwald, 2022*; *Aglinskas and Fairhall, 2023*; *Hauptman et al., 2025*). By contrast, parts of the visual system (e.g. fusiform face area [FFA]) that respond preferentially to animates do so primarily for images (*Kanwisher et al., 1997*; *Grill-Spector et al., 2004*; *Noppeney et al., 2006*; *Mahon et al., 2009*; *Konkle and Caramazza, 2013*; *Connolly et al., 2016*; see *Bi et al., 2016*, for a review). We hypothesized that the PC represents causal knowledge relevant to animates and tested the prediction that it would be activated during causal inferences about illness, which rely on such knowledge (preregistration: https://osf.io/6pnqg).

We also compared neural responses to causal inferences about the body (i.e. illness) and inferences about the mind (i.e. mental states). Both types of inferences are about animate entities, and some developmental work suggests that children use the same set of causal principles to think about bodies and minds (*Carey, 1985*; *Carey, 1988*). Other evidence suggests that by early childhood, young children have distinct causal knowledge about the body and the mind (*Springer and Keil, 1991*; *Callanan and Oakes, 1992*; *Wellman and Gelman, 1992*; *Inagaki and Hatano, 1993*; *Inagaki and Hatano, 2004*; *Keil, 1994*; *Hickling and Wellman, 2001*; *Medin et al., 2010*). For instance, preschoolers are more likely to view illness as a consequence of biological causes, such as contagion, rather than psychological causes, such as malicious intent (*Springer and Ruckel, 1992*; *Raman and Winer, 2004*; see also *Legare and Gelman, 2008*). The neural relationship between inferences about bodies and minds has not been fully described. The 'mentalizing network', including the PC, is engaged when people reason about agents' beliefs (*Saxe and Kanwisher, 2003*; *Saxe et al., 2006*; *Saxe and Powell, 2006*; *Dodell-Feder et al., 2011*; *Dufour et al., 2013*). We localized this network in individual participants and measured its neuroanatomical relationship to the network activated by illness inferences.

An alternative hypothesis is that domain-general neural mechanisms, separate from semantic networks, support causal inferences across domains. Children and adults make causal inferences across a wide range of domains and use similar cognitive principles (e.g. 'screening off') when doing so (e.g. *Saxe and Carey, 2006*; *Tenenbaum et al., 2007*; *Carey, 2011*; *Cheng and Novick, 1992*; *Waldmann and Holyoak, 1992*; *Pearl, 2000*; *Gopnik et al., 2001*; *Steyvers et al., 2003*; *Gopnik et al., 2004*; *Schulz and Gopnik, 2004*; *Rehder and Burnett, 2005*; *Lagnado et al., 2007*; *Rottman and Hastie, 2014*; *Davis and Rehder, 2020*). Prior neuroscience work has hypothesized that the frontotemporal language network may support a broad range of causal inferences during comprehension (*Kuperberg et al., 2006*; *Mason and Just, 2011*; *Prat et al., 2011*; see also *Spelke, 2003*; *Spelke, 2022*; *Pinker, 2003*). Alternatively, causal inference could depend on frontoparietal mechanisms that also support other types of reasoning, such as logical deduction (*Goldvarg and Johnson-Laird, 2001*; *Barbey and Patterson, 2011*; *Khemlani et al., 2014*; *Operskalski and Barbey, 2017*). Finally, it has been suggested that causal inferences are supported by a dedicated 'causal engine' in prefrontal cortex that supports all and only causal inferences across domains (*Pramod et al., 2023*). We tested these alternative hypotheses in the specific case of implicit causal inferences that unfold naturally during language comprehension (*Black and Bern, 1981*; *Keenan et al., 1984*; *Trabasso and Sperry, 1985*; *Myers et al., 1987*; *Duffy et al., 1990*).

Most prior studies investigating causal inference used explicit causality judgment tasks (*Ferstl and von Cramon, 2001*; *Satpute et al., 2005*; *Fugelsang and Dunbar, 2005*; *Kuperberg et al., 2006*; *Fenker et al., 2010*; *Kranjec et al., 2012*; *Pramod et al., 2023*). For example, *Kuperberg et al.,*

*2006* asked participants to rate the causal relatedness of three-sentence stories and observed higher responses to causally related stories in left frontotemporal cortex. Studies of implicit causal inference report frontotemporal and frontoparietal responses (*Chow et al., 2008*; *Mason and Just, 2011*; *Prat et al., 2011*). Across these prior studies, no consistent neural signature of causal inference has emerged. Importantly, in many studies, causal trials were more difficult, and/or linguistic variables were not matched across causal and noncausal conditions. As a result, some of the observed effects may reflect linguistic or executive load. In addition, almost no prior studies localized language or logical reasoning networks in individual participants, making it difficult to assess the involvement of these systems (e.g. *Fedorenko et al., 2010*; *Monti et al., 2009*; *Pramod et al., 2023*). Most prior work also did not distinguish between causal inferences about different semantic domains known to depend on partially distinct neural networks, e.g., biological, mechanical, or mental state inferences (cf. *Mason and Just, 2011*; *Pramod et al., 2023*). If such inferences recruit partially distinct neural systems, their neural signatures might have been missed.

In the current experiment, participants read two-sentence vignettes (e.g. 'Hugh sat by sneezing passengers on the subway. Now he has a case of COVID.'). The first sentence described a potential cause and the second sentence a potential effect. Such causally connected sentences arise frequently in naturalistic discourse (*Singer, 1994*; *Graesser et al., 1994*). Participants performed a covert task of detecting 'magical' catch trial vignettes that encouraged them to attend to the meaning of the critical vignettes while reading as naturally as possible. We chose an orthogonal foil detection task rather than an explicit causal judgment task to investigate automatic causal inferences during reading and to unconfound such processing as much as possible from explicit decision-making processes. Analogous foil detection paradigms have been used to study sentence processing and word recognition (e.g. *Pallier et al., 2011*; *Dehaene-Lambertz et al., 2018*).

Causal inferences about illness were compared to two control conditions: (i) causal inferences about mechanical breakdown (e.g. 'Jake dropped all of his things on the subway. Now he has a shattered phone.') and (ii) illness-related language that was not causally connected (e.g. 'Lynn dropped all of her things on the subway. Now she has a case of COVID.'). This combination of control conditions allowed us to test jointly for sensitivity to content domain and causality. In other words, this design enabled us to test the hypothesis that causal inferences about illness recruit the animacy semantic network. Critically, all vignettes, including mechanical ones, described events involving people, such that responses to causal inferences about illness in the animacy semantic network could not be explained by the presence of animate agents. As a further control, we included the number of people in each vignette as a covariate of no interest in our fMRI analysis. Noncausal vignettes were constructed by shuffling causes/effects across conditions and were therefore matched to the causal vignettes in linguistic content. A separate group of participants rated the causal relatedness of all vignettes prior to the experiment. In addition to the main causal inference experiment, we also localized language, logical reasoning, and mentalizing networks in each participant. Following prior work, we predicted that the neural systems that support causal inference would exhibit increased activity during such inferences. Thus, our primary neural prediction was that animacy-responsive PC would respond more to causal inferences about illness compared to all other control conditions. We also used multivariate methods to investigate differences between conditions.

## Results

### Behavioral results

Accuracy on the magic detection task was at ceiling ($M$=97.9% ± 2.2 SD), and there were no significant differences across the four main experimental conditions (*Illness-Causal*, *Mechanical-Causal*, *Noncausal-Illness First, Noncausal-Mechanical First*), $F_{(3,57)}$ = 2.39, p=0.08. A one-way repeated-measures ANOVA evaluating response time revealed a main effect of condition, $F_{(3,57)}$ = 32.63, p<0.001, whereby participants were faster on *Illness-Causal* trials ($M$=4.73 ± 0.81 SD) compared to *Noncausal-Illness First* ($M$=5.33 s±0.85 SD) and *Noncausal-Mechanical First* ($M$=5.27 s±0.89 SD) trials. There were no differences in response time between the *Mechanical-Causal* condition ($M$=5.15 s±0.88 SD) and any other conditions. Performance on the localizer tasks was similar to previously reported studies that used these paradigms (see Appendix 3 for full behavioral results).

## Inferring illness causes recruits animacy-responsive PC

We found distinctly localized neural responses to causal inferences about illness relative to both mechanical causal inferences and noncausal vignettes. A bilateral PC region previously implicated in thinking about animate entities (i.e. people and animals) responded preferentially to causal inferences about illness over both mechanical causal inferences and causally unrelated sentences in whole-cortex analysis (p<0.05, corrected for multiple comparisons; *Figure 1C*) and in individual-subject overlap maps (*Figure 1—figure supplement 1; Figure 1—figure supplement 2*). PC responses during illness inferences overlapped with previously reported responses to people-related concepts (*Fairhall and Caramazza, 2013b*; *Figure 1—figure supplement 3*).

Relative to illness inferences and noncausal vignettes, inferring the causes of mechanical breakdown in inanimate entities activated bilateral anterior parahippocampal regions (i.e. anterior PPA), suggesting a double dissociation between illness and mechanical inferences (*Figure 2*; *Epstein and Kanwisher, 1998*; *Weiner et al., 2018*). This anterior PPA region is engaged during memory/verbal tasks about physical spaces (*Baldassano et al., 2013*; *Fairhall et al., 2014*; *Silson et al., 2019*; *Steel et al., 2021*; *Häusler et al., 2022*; *Hauptman et al., 2025*).

In individual-subject functional ROI (fROI) analysis (leave-one-run-out), we similarly found that inferring illness causes activated the PC more than inferring causes of mechanical breakdown (repeated-measures ANOVA, condition (*Illness-Causal, Mechanical-Causal*) × hemisphere (left, right): main effect of condition, $F_{(1,19)} = 19.18$, p<0.001, main effect of hemisphere, $F_{(1,19)} = 0.3$, p=0.59, condition × hemisphere interaction, $F_{(1,19)} = 27.48$, p < 0.001; *Figure 1A*). This effect was larger in the left than in the right PC (paired samples t-tests; left PC: $t_{(19)} = 5.36$, p<0.001, right PC: $t_{(19)} = 2.27$, p=0.04). Illness inferences also activated the PC more than illness-related language that was not causally connected (repeated-measures ANOVA, condition (*Illness-Causal, Noncausal-Illness First*) × hemisphere (left, right): main effect of condition, $F_{(1,19)} = 4.66$, p=0.04, main effect of hemisphere, $F_{(1,19)} = 2.51$, p=0.13, condition × hemisphere interaction, $F_{(1,19)} = 8.07$, p=0.01; repeated-measures ANOVA, condition (*Illness-Causal, Noncausal-Mechanical First*) × hemisphere left, right: main effect of condition, $F_{(1,19)} = 4.38$, p=0.05; main effect of hemisphere, $F_{(1,19)} = 1.17$, p = 0.29; condition × hemisphere interaction, $F_{(1,19)} = 17.89$, p<0.001; *Figure 1A*). Both effects were significant only in the left PC (paired samples t-tests; *Illness-Causal* vs. *Noncausal-Illness First*, left PC: $t_{(19)} = 2.77$, p=0.01, right PC: $t_{(19)} = 1.28$, p=0.22; *Illness-Causal* vs. *Noncausal-Mechanical First,* left PC: $t_{(19)} = 3.21$, p=0.005, right PC: $t_{(19)} = 0.5$, p = 0.62).

We also observed increased activity for illness inferences compared to mechanical inferences in the temporoparietal junction (TPJ) (leave-one-run-out individual-subject fROI analysis; repeated-measures ANOVA, condition (*Illness-Causal, Mechanical-Causal*) × hemisphere (left, right): main effect of condition, $F_{(1,19)} = 5.33$, p=0.03, main effect of hemisphere, $F_{(1,19)} = 1.02$, p=0.33, condition × hemisphere interaction, $F_{(1,19)} = 4.24$, p=0.05; *Figure 1—figure supplements 4 and 5*). This effect was significant only in the left TPJ (paired samples t-tests; left TPJ: $t_{(19)} = 2.64$, p=0.02, right TPJ: $t_{(19)} = 1.13$, p=0.27). Unlike the PC, the TPJ did not show a preference for illness inferences compared to illness-related language that was not causally connected (repeated-measures ANOVA, condition (*Illness-Causal, Noncausal-Illness First*) × hemisphere (left, right): main effect of condition, $F_{(1,19)} = 0.006$, p=0.94, main effect of hemisphere, $F_{(1,19)} = 2.19$, p=0.16, condition × hemisphere interaction, $F_{(1,19)} = 1.27$, p=0.27; repeated-measures ANOVA, condition (*Illness-Causal, Noncausal-Mechanical First*) × hemisphere (left, right): main effect of condition, $F_{(1,19)} = 0.73$, p=0.41; main effect of hemisphere, $F_{(1,19)} = 1.24$, p=0.28; condition × hemisphere interaction, $F_{(1,19)} = 3.34$, p=0.08; *Figure 1—figure supplements 4 and 5*).

In contrast to animacy-responsive PC, the anterior PPA showed the opposite pattern, responding more to mechanical inferences than illness inferences (leave-one-run-out individual-subject fROI analysis; repeated-measures ANOVA, condition (*Mechanical-Causal, Illness-Causal*) × hemisphere (left, right): main effect of condition, $F_{(1,19)} = 17.93$, p<0.001, main effect of hemisphere, $F_{(1,19)} = 1.33$, p=0.26, condition × hemisphere interaction, $F_{(1,19)} = 7.8$, p=0.01; *Figure 2*). This effect was significant only in the left anterior PPA (paired samples t-tests; left anterior PPA: $t_{(19)} = 4$, p<0.001, right anterior PPA: $t_{(19)} = 1.88$, p=0.08). The anterior PPA also showed a preference for mechanical inferences compared to mechanical-related language that was not causally connected (repeated-measures ANOVA, condition (*Mechanical-Causal, Noncausal-Illness First*) × hemisphere (left, right): main effect of condition, $F_{(1,19)} = 14.81$, p=0.001, main effect of hemisphere, $F_{(1,19)} = 1.81$, p=0.2, condition × hemisphere interaction, $F_{(1,19)} = 7.35$, p=0.01; repeated-measures ANOVA, condition (*Mechanical-Causal,*

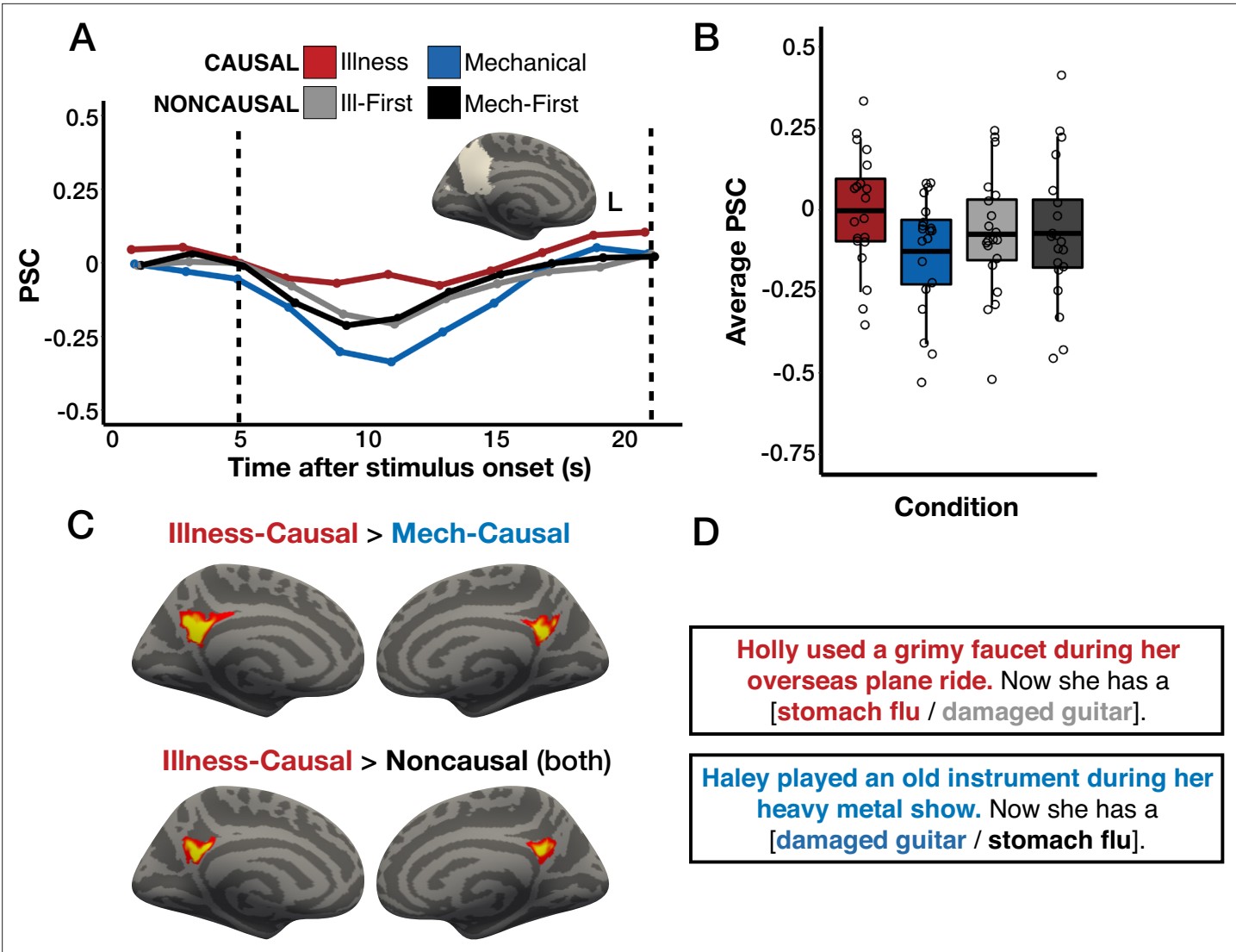

**Figure 1.** Responses to illness inferences in the precuneus (PC). (**A**) Percent signal change (PSC) for each condition among the top 5% *Illness-Causal>Mechanical-Causal* vertices in a left PC search space (*Dufour et al., 2013*) in individual participants, established via a leave-one-run-out analysis. (**B**) Average PSC in the critical window (marked by dotted lines in A) across participants. The horizontal line within each boxplot indicates the overall mean. (**C**) Whole-cortex results (one-tailed) for *Illness-Causal>Mechanical-Causal* and *Illness-Causal>Noncausal* (both versions of noncausal vignettes), corrected for multiple comparisons (p<0.05 family-wise error rate [FWER], cluster-forming threshold p<0.01 uncorrected). Vertices are color-coded on a scale from p=0.01 to p=0.00001. (**D**) Example stimuli. 'Magical' catch trials similar in meaning and structure (e.g. 'Sadie forgot to wash her face after she ran in the heat. Now she has a cucumber nose.') enabled the use of a semantic 'magic detection' task.

The online version of this article includes the following figure supplement(s) for figure 1:

**Figure supplement 1.** Group overlap in univariate contrasts comparing causal (*Illness-Causal*, *Mechanical-Causal*) and noncausal conditions (*Noncausal-Illness First + Noncausal-Mechanical First*) in the precuneus (PC), winner-take-all approach.

**Figure supplement 2.** Group overlap in univariate contrasts comparing causal (*Illness-Causal*, *Mechanical-Causal*) and noncausal conditions (*Noncausal-Illness First + Noncausal-Mechanical First*) in the precuneus (PC).

**Figure supplement 3.** Overlap between left precuneus (PC) responses to illness inferences in the current study and people-related stimuli in a separate study (*Fairhall and Caramazza, 2013b*).

**Figure supplement 4.** Responses to illness inferences in bilateral precuneus (PC) and temporoparietal junction (TPJ).

**Figure supplement 5.** Subject dispersion data for responses to illness inferences in bilateral precuneus (PC) and temporoparietal junction (TPJ) (see *Figure 1—figure supplement 4*).

**Figure supplement 6.** Percent signal change (PSC) for each condition among the top 5% Illness-Causal>Mechanical-Causal vertices in a left precuneus (PC) search space (*Dufour et al., 2013*) in individual participants, established via a leave-one-run-out analysis.

*Figure 1 continued on next page*

*Figure 1 continued*

**Figure supplement 7.** Functional localization of language, logical reasoning, and mentalizing networks (see *Monti et al., 2009*; *Fedorenko et al., 2010*; *Dodell-Feder et al., 2011*; *Liu et al., 2020*).

**Figure supplement 8.** Full whole-cortex univariate results.

**Figure supplement 9.** Comparison of whole-cortex results for number of people in each vignette (left) and illness inferences (right) from the same generalized linear model (GLM).

**Figure supplement 10.** Searchlight MVPA group maps.

**Figure supplement 11.** Subject dispersion data for individual-subject MVPA performed in functional ROIs (fROIs).

**Figure supplement 12.** Subject dispersion data for individual-subject MVPA performed in functional ROIs (fROIs).

**Figure supplement 13.** Responses to illness inferences in the fusiform face area (FFA).

**Figure supplement 14.** Comparison of mentalizing localizers used in previous work and in the current study, in three pilot participants.

*Noncausal-Mechanical First*) × hemisphere (left, right): main effect of condition, $F_{(1,19)} = 11.31$, p=0.003; main effect of hemisphere, $F_{(1,19)} = 3.34$, p=0.08; condition × hemisphere interaction, $F_{(1,19)} = 4$, p=0.06; *Figure 2*). Similar to the PC, both effects were larger in the left than in the right hemisphere (post hoc paired samples t-tests; *Illness-Causal* vs. *Noncausal-Illness First*, left anterior PPA: $t_{(19)} = 3.85$, p=0.001, right anterior PPA: $t_{(19)} = 2.22$, p=0.04; *Illness-Causal* vs. *Noncausal-Mechanical First,* left anterior PPA: $t_{(19)} = 3.59$, p=0.002, right anterior PPA: $t_{(19)} = 1.19$, p=0.25).

In summary, we found distinctly localized responses to illness and mechanical causal inferences. Inferring illness causes preferentially recruited the animacy semantic network, particularly the PC.

## Illness inferences and mental state inferences elicit spatially dissociable responses

Illness inferences and mental state inferences elicited spatially dissociable responses. In whole-cortex analysis, illness inferences recruited the PC bilaterally, with larger responses observed in the left hemisphere (*Figure 1*, see also fROI analysis showing left-lateralization above). By contrast, and in accordance with prior work (e.g. *Saxe and Kanwisher, 2003*), mental state inferences recruited a broader

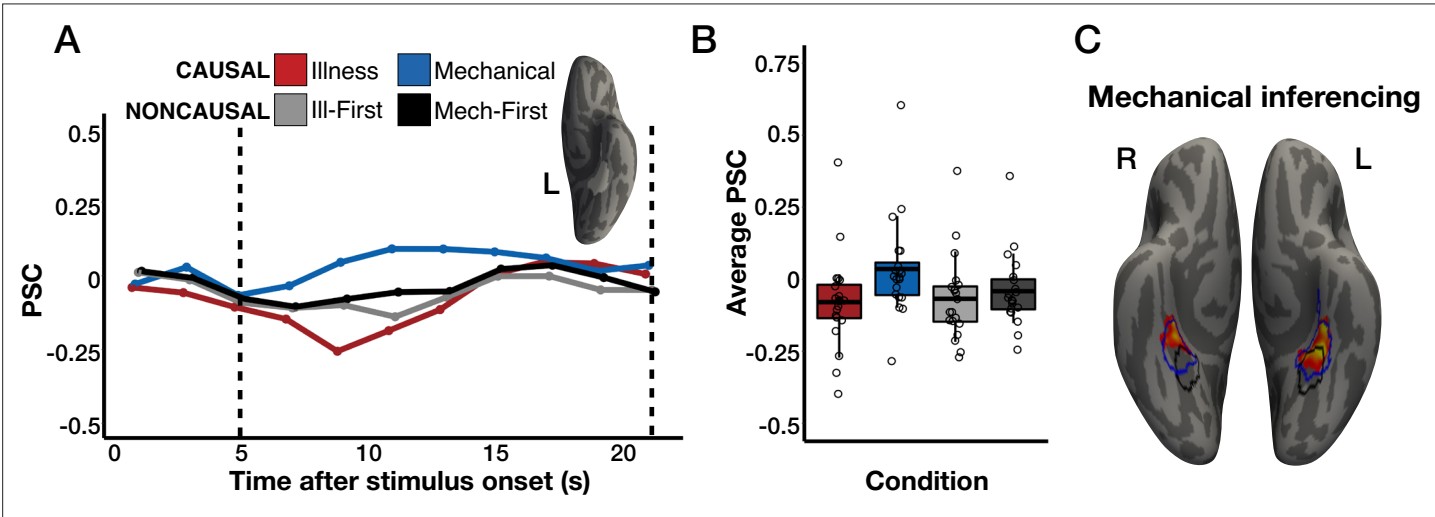

**Figure 2.** Responses to mechanical inferences in anterior parahippocampal regions (anterior PPA). (**A**) Percent signal change (PSC) for each condition among the top 5% *Mechanical-Causal>Illness-Causal* vertices in a left anterior PPA search space (*Hauptman et al., 2025*) in individual participants, established via a leave-one-run-out analysis. (**B**) Average PSC in the critical window (marked by dotted lines in A) across participants. The horizontal line within each boxplot indicates the overall mean. (**C**) The intersection of two whole-cortex contrasts (one-tailed), *Mechanical-Causal>Illness-Causal* and *Mechanical-Causal>Noncausal* that are corrected for multiple comparisons (p<0.05 family-wise error rate [FWER], cluster-forming threshold p<0.01 uncorrected). Vertices are color-coded on a scale from p=0.01 to p=0.00001. Similar to PC responses to illness inferences, anterior PPA is the only region to emerge across both mechanical inference contrasts. The average PPA location from a separate study involving perceptual place stimuli (*Weiner et al., 2018*) is overlaid in black. The average PPA location from a separate study involving verbal place stimuli (*Hauptman et al., 2025*) is overlaid in blue.

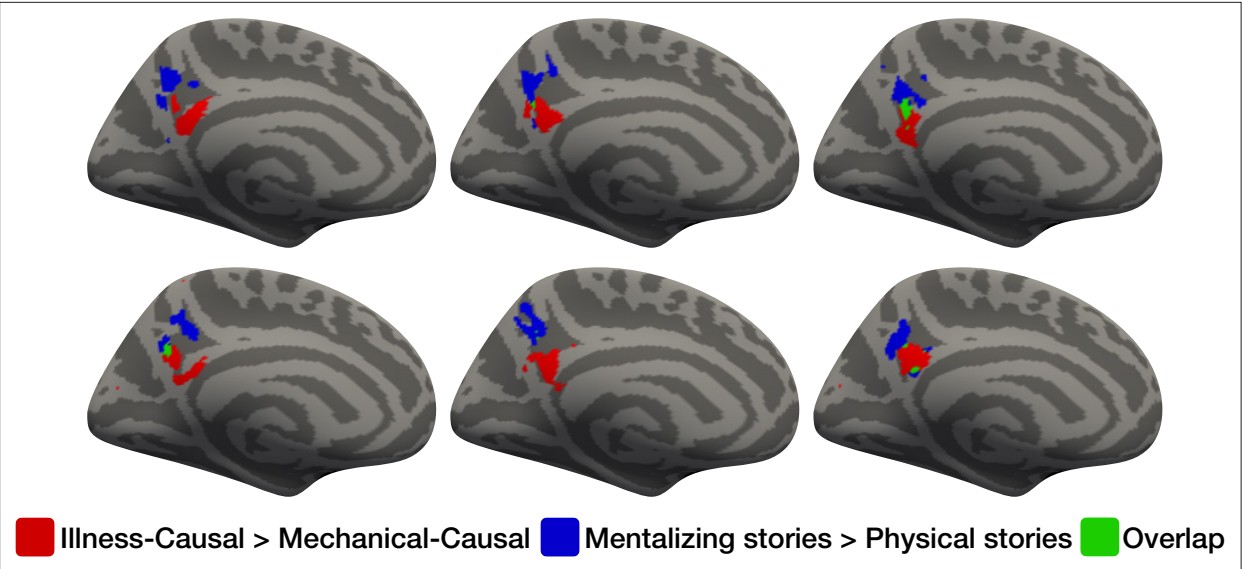

Illness-Causal > Mechanical-Causal ▮ Mentalizing stories > Physical stories ▮ Overlap ▮

**Figure 3.** Spatial dissociation between univariate responses to illness inferences and mental state inferences in the precuneus (PC). The left medial surface of six individual participants were selected for visualization purposes. The locations of the top 10% most responsive vertices to *Illness-Causal>Mechanical-Causal* in a PC search space (*Dufour et al., 2013*) are shown in red. The locations of the top 10% most responsive vertices to *mentalizing stories >physical stories* (mentalizing localizer) in the same PC search space are shown in blue. Overlapping vertices are shown in green.

The online version of this article includes the following figure supplement(s) for figure 3:

**Figure supplement 1.** Spatial dissociation between responses to illness inferences and mental state inferences in left precuneus (PC).

network, including not only bilateral PC, but also bilateral TPJ, superior temporal sulcus, and medial and superior prefrontal cortex (*Figure 1—figure supplement 7*).

Within the left PC, responses to illness inferences were located ventrally to mental state inference responses (*Figure 3*, *Figure 3—figure supplement 1*). The z-coordinates of individual-subject activation peaks for illness inferences and mental state inferences were significantly different (repeated-measures ANOVA, $F_{(1,19)}$ = 13.52, p=0.002). In addition, the size of the illness inference effect (*Illness-Causal >Mechanical-Causal*) was larger in illness-responsive vertices (leave-one-run-out individual-subject fROI analysis) than in mentalizing-responsive vertices in the left PC (individual-subject fROI analysis; repeated-measures ANOVA, $F_{(1,19)}$ = 24.72, p<0.001, *Figure 1—figure supplements 4 and 5*). These results suggest that illness inferences and mental state inferences are carried out by neighboring but partially distinct subsets of the PC.

## No univariate evidence for domain-general responses to implicit causal inference

Prior neuroscience studies hypothesizing the existence of a domain-general 'causal engine' have predicted that the language network and/or domain-general executive systems (e.g. the logic network) should show elevated activity during causal inference across domains. In the current study, neither the language nor the logic network exhibited elevated neural responses during causal inferences relative to linguistically matched sentence pairs that were not causally connected. Language regions in fronto-temporal cortex responded more to noncausal than causal vignettes (frontal search space: repeated-measures ANOVA, $F_{(1,19)}$ = 23.91, p<0.001; temporal search space: repeated-measures ANOVA, $F_{(1,19)}$ = 4.31, p=0.05; *Figure 4*, *Figure 4—figure supplement 1*). The logic network likewise responded marginally more to noncausal vignettes, likely reflecting greater difficulty associated with integrating unrelated sentences (repeated-measures ANOVA, $F_{(1,19)}$ = 3.88, p=0.07; *Figure 4*).

In whole-cortex univariate analysis, no shared regions responded more to causal than noncausal vignettes across domains. Two whole-cortex univariate contrasts comparing causal and noncausal conditions (*Illness-Causal>Noncausal-Mechanical First*, *Mechanical-Causal>Noncausal-Mechanical First*) revealed increased activity for the noncausal condition in bilateral prefrontal cortex. The same prefrontal areas that responded more to noncausal than causal stimuli also responded more when

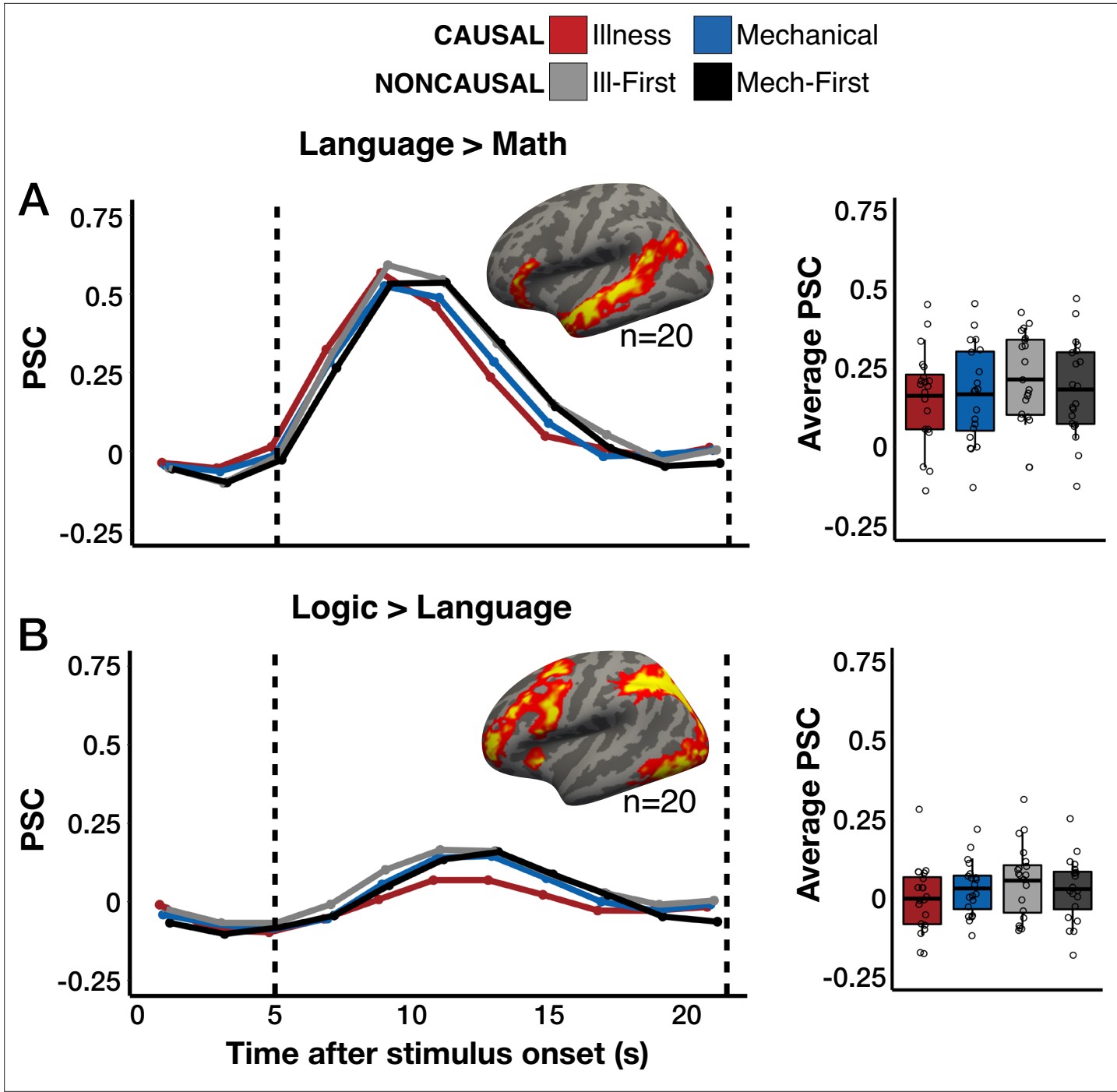

**Figure 4.** Individual-subject analysis of language- and logic-responsive vertices. (**A**) Percent signal change (PSC) for each condition among the top 5% most language-responsive vertices (language>*math*) in a temporal language network search space (*Fedorenko et al., 2010*). Results from a frontal language search space (*Fedorenko et al., 2010*) can be found in *Figure 4—figure supplement 1*. (**B**) PSC among the top 5% most logic-responsive vertices (logic>*language*) in a logic network search space (*Liu et al., 2020*). Group maps for each contrast of interest (one-tailed) are corrected for multiple comparisons (p<0.05 family-wise error rate [FWER], cluster-forming threshold p<0.01 uncorrected). Vertices are color-coded on a scale from p=0.01 to p=0.00001. Boxplots display average PSC in the critical window (marked by dotted lines) across participants. The horizontal line within each boxplot indicates the overall mean.

The online version of this article includes the following figure supplement(s) for figure 4:

**Figure supplement 1.** Responses to causal inference in the language network.

participants were slower to complete the task, suggesting that these responses reflect a nonspecific difficulty effect (*Figure 1—figure supplement 8*).

In summary, none of the predicted networks nor any regions across the whole cortex exhibited the predicted domain-general causal inference pattern, i.e., larger responses to all causal than all noncausal vignettes. These results suggest that implicit causal inferences, which draw upon a person's existing knowledge of relevant causes and effects, do not depend on domain-general neural mechanisms. These results leave open the possibility that domain-general systems support the explicit search for causal connections (see Discussion section).

## Multivariate analysis

In searchlight MVPA performed across the whole cortex, illness inferences and mechanical inferences produced spatially distinguishable neural patterns in the left PC extending dorsally into the superior parietal lobule, as well as in left anterior PPA and lateral occipitotemporal cortex. A whole-cortex searchlight analysis that tested whether each causal condition could be decoded from each noncausal condition found no shared regions that exhibited significant decoding across all causal vs. noncausal comparisons (*Figure 1—figure supplement 10*).

In individual-subject fROI decoding analyses, illness inferences and mechanical inferences produced spatially distinguishable neural patterns in the left PC, right PC, and left TPJ, as well as in language and logic networks (see *Figure 1—figure supplement 12*, *Supplementary file 2* for full results). Note that these decoding results must be interpreted in light of the significant univariate differences observed across conditions that are reported above. Linear classifiers are highly sensitive to univariate differences (*Coutanche, 2013*; *Kragel et al., 2012*; *Hebart and Baker, 2018*; *Woolgar et al., 2014*; *Davis et al., 2014*; *Pakravan et al., 2022*). Successful decoding may be driven by univariate differences in the predicted direction (e.g. causal>noncausal) or in the opposite direction (e.g. noncausal>causal). In particular, given that both the language and the logic networks exhibited higher univariate responses to noncausal compared to causal vignettes, decoding results observed in these networks may be driven by univariate differences.

## Discussion
### Causal knowledge is embedded in high-level semantic networks

We find that a semantic network previously implicated in thinking about animates, particularly the (PC), is preferentially engaged when people infer causes of illness compared to when they infer causes of mechanical breakdown or read causally unconnected sentences containing illness-related language. By contrast, mechanical inferences activate an anterior parahippocampal region previously implicated in thinking about and remembering places (*Baldassano et al., 2013*; *Fairhall et al., 2014*; *Silson et al., 2019*; *Steel et al., 2021*; *Häusler et al., 2022*; *Hauptman et al., 2025*). This finding points to a neural double dissociation between biological and mechanical causal knowledge.

Previous work has implicated the PC in the representation of animate entities, i.e., people and animals (*Fairhall and Caramazza, 2013a*; *Fairhall et al., 2014*; *Peer et al., 2015*; *Wang et al., 2016*; *Silson et al., 2019*; *Rabini et al., 2021*; *Deen and Freiwald, 2022*; *Aglinskas and Fairhall, 2023*; *Hauptman et al., 2025*). Here, we show that the PC exhibits sensitivity to causal inferences about biological processes specific to animates, such as illness. These findings are consistent with our preregistered hypotheses and suggest that causal knowledge about animate and inanimate entities is distributed across multiple distinct semantic networks. Further, our results suggest that the animacy semantic network supports biological causal knowledge. Future work should test whether the animacy network is sensitive to causal information beyond illness, including about growth, nourishment, and death. We hypothesize that changes in biological causal knowledge during development, as well as cultural expertise in causal reasoning about illness (e.g. medical expertise), influence activity in the animacy network (*Legare et al., 2012*; *Norman et al., 2009*).

Our findings are consistent with prior evidence from naturalistic paradigms showing that the PC is sensitive to discourse-level processes across sentences (e.g. *Hasson et al., 2008*; *Lerner et al., 2011*; *Lee and Chen, 2022*). We hypothesize that PC responses observed during naturalistic narrative comprehension are driven by causal inferences about animate agents, who are often the focus of narratives. Likewise, PC involvement in episodic memory could be related to animacy-related inferential

processes (*DiNicola et al., 2020*; *Ritchey and Cooper, 2020*). Future work can test this hypothesis by comparing causal inferences about animate and inanimate entities in naturalistic contexts, such as films and verbal narratives (see *Chen and Bornstein, 2024*, for a review on causal inference in narrative comprehension).

We find that neural responses during inferences about biological and mental properties of animates are linked yet separable. Inferring illness causes recruits neural circuits that are adjacent to but distinct from responses to mental state inferences in the PC (*Saxe and Kanwisher, 2003*; *Saxe et al., 2006*). Even young children provide different causal explanations for biological vs. psychological processes (*Springer and Keil, 1991*; *Callanan and Oakes, 1992*; *Wellman and Gelman, 1992*; *Inagaki and Hatano, 1993*; *Inagaki and Hatano, 2004*; *Keil, 1994*; *Hickling and Wellman, 2001*; *Medin et al., 2010*; cf. *Carey, 1985*; see also *Medin and Atran, 2004*). For example, when asked why blood flows to different parts of the body, 6-year-old endorse explanations referring to bodily function, e.g., 'because it provides energy to the body,' and not to mental states, e.g., 'because we want it to flow' (*Inagaki and Hatano, 1993*). At the same time, animate entities have a dual nature: they have both bodies and minds (*Opfer and Gelman, 2011*; *Spelke, 2022*). The current findings point to the existence of distinct but related neural systems for biological and mentalistic knowledge.

Our neuroimaging findings are consistent with evidence from developmental psychology suggesting that causal knowledge is central to human concepts starting early in development (*Keil, 1992*; *Wellman and Gelman, 1992*; *Hatano and Inagaki, 1994*; *Springer and Keil, 1991*; *Simons and Keil, 1995*; *Atran, 1998*; *Keil et al., 1999*; *Coley et al., 2002*; *Medin and Atran, 2004*). According to the 'intuitive theories' account, semantic knowledge is organized into causal frameworks that serve as 'grammars for causal inference' (*Tenenbaum et al., 2007*; *Wellman and Gelman, 1992*; *Gopnik and Meltzoff, 1997*; *Gopnik and Wellman, 2012*; *Gerstenberg and Tenenbaum, 2017*; see also *Boyer, 1995*; *Barrett et al., 2007*; *Cosmides and Tooby, 2013*; *Bender et al., 2017*). For example, preschoolers intuit that animates but not inanimate objects get sick and need nourishment to grow and live (e.g. *Rosengren et al., 1991*; *Kalish, 1996*; *Gutheil et al., 1998*; *Raman and Gelman, 2005*; see *Inagaki and Hatano, 2004*; *Opfer and Gelman, 2011*, for reviews). The present results suggest that such knowledge is encoded in high-level semantic brain networks. By contrast, we failed to find sensitivity to causal inference in portions of the ventral stream previously associated with the perception of animate agents (see Appendix 4, *Figure 1—figure supplement 13* for details). Sensitivity to causal information may be a distinguishing characteristic of high-level, amodal semantic networks, as opposed to perceptual regions that are activated during semantic tasks (e.g. *Martin and Chao, 2001*; *Thompson-Schill, 2003*; *Barsalou et al., 2003*; *Binder and Desai, 2011*; *Bi, 2021*).

## No evidence for domain-general neural responses during implicit causal inference

In the current study, participants read two sentence vignettes that either elicited causal inferences or were not causally connected. No brain regions responded more to causal inferences across domains compared to noncausal vignettes in this task. The language network responded more to noncausal than causal vignettes, possibly due to greater difficulty associated with processing the meaning of a sentence that does not follow from the prior context. Prior studies find that the language network is specialized primarily for sentence-internal processing (*Fedorenko and Varley, 2016*; *Jacoby and Fedorenko, 2020*; *Blank and Fedorenko, 2020*) and patients with agrammatic aphasia can make causal inferences about pictorial stimuli (*Varley and Siegal, 2000*; *Varley, 2014*). Together, these results suggest that the language system itself is unlikely to support causal inference. Rather, during language comprehension, the language system interacts with semantic networks to enable causal inference (*Simony et al., 2016*; *Yeshurun et al., 2017*; *Chang et al., 2022*). Notably, in the current study, responses to causal inference in semantic networks were stronger in the left hemisphere. The left lateralization of such responses may enable efficient interfacing with the language system during comprehension.

We also failed to find evidence for the claim that the frontoparietal logical reasoning network, a domain-general executive system, supports implicit causal inferences. By contrast, the frontoparietal network responded more to noncausal than causal vignettes. Finally, we failed to observe elevated responses to causal inference across domains anywhere in the brain in whole-cortex analysis. A large swath of prefrontal cortex responded more to one noncausal condition (*Noncausal-Mechanical First*)

compared to both causal conditions. The same prefrontal regions also exhibited increased activity when participants were slower to respond to the task. Thus, this 'reverse causality effect' likely reflects processing demands rather than causal inference per se. An alternative interpretation of the elevated prefrontal activity observed for one of the noncausal conditions is that it reflects the effortful search for a causal connection between sentences when such a connection is difficult to find. This interpretation would suggest that domain-general executive mechanisms become engaged when causal inferences are effortful and explicit. By contrast, semantic systems are engaged when we implicitly infer a known causal relationship.

Causal inferences are a highly varied class, and domain-general systems likely play an important role in many causal inferences not tested in the current study. The vignettes used in the current study stipulate illness causes, allowing participants to reason from causes to effects. By contrast, illness reasoning performed by medical experts proceeds from effects to causes and can involve searching for potential causes within highly complicated and interconnected causal systems (*Schmidt et al., 1990*; *Norman et al., 2009*; *Meder and Mayrhofer, 2017*). The discovery of novel causal relationships (e.g. 'blicket detectors'; *Gopnik et al., 2001*) and the identification of complex causes, even in the case of illness, may depend in part on domain-general neural mechanisms. The present results suggest, however, that causal knowledge is embedded within high-level semantic systems, and that biological causal knowledge is embedded with a semantic system relevant to animacy.

## Materials and methods
### Open science practices
The methods and analysis of this experiment were preregistered prior to data collection (https://osf.io/6pnqg).

### Participants
Twenty adults (7 women, 13 men, 25–37 years of age, $M$=28.7 years±3.2 SD) participated in the study. Participants either had or were pursuing graduate degrees ($M$=8.8 years of post-secondary education). Two additional participants were excluded from the final dataset due to excessive head motion (>2 mm) and an image artifact. One participant in the final dataset exhibited excessive head motion (>2 mm) during one run of the language/logic localizer task that was excluded from analysis. All participants were screened for cognitive and neurological disabilities (self-report). Participants gave written informed consent and were compensated $30 per hour. The study was reviewed and approved by the Johns Hopkins Medicine Institutional Review Boards (IRB00270868).

### Causal inference experiment
#### Stimuli
Participants read two-sentence vignettes in four conditions, two causal and two noncausal (*Figure 1D*). Each vignette focused on a single agent, specified by a proper name in the initial sentence and by a pronoun in the second sentence. The first sentence described something the agent did or experienced and served as the potential cause. The second sentence described the potential effect (e.g. 'Kelly shared plastic toys with a sick toddler at her preschool. Now she has a case of chickenpox.'). *Illness-Causal* vignettes elicited inferences about biological causes of illness, including pathogen transmission, exposure to environmental toxins, and genetic mutations (see *Supplementary file 1* for a full list of the types of illnesses included in our stimuli). *Mechanical-Causal* vignettes elicited inferences about physical causes of structural damage to personally valuable inanimate objects (e.g. houses, jewelry). Two noncausal conditions used the same sentences as in the *Illness-Causal* and *Mechanical-Causal* conditions but in a shuffled order: illness cause with mechanical effect (*Noncausal-Illness First*) or mechanical cause with illness effect (*Noncausal-Mechanical First*). Explicit causality judgments collected from a separate group of online participants ($n$=26) verified that both causal conditions *Illness-Causal*, *Mechanical-Causal* were more causally related than both noncausal conditions, $t(25) = 36.97$, p<0.001. In addition, *Illness-Causal* and *Mechanical-Causal* items received equally high causality ratings, $t(25) = –0.64$, p=0.53 (see Appendix 1 for details).

*Illness-Causal* and *Mechanical-Causal* vignettes were constructed in pairs, such that each member of a given pair shared parallel or near-parallel phrase structure. All conditions were also matched

(pairwise t-tests, all ps>0.3, no statistical correction) on multiple linguistic variables known to modulate neural activity in language regions (e.g. *Pallier et al., 2011*; *Shain et al., 2020*). These included number of characters, number of words, average number of characters per word, average word frequency, average bigram surprisal (Google Books Ngram Viewer, https://books.google.com/ngrams/), and average syntactic dependency length (Stanford Parser; *Marneffe et al., 2006*). Word frequency was calculated as the negative log of a word's frequency in the Google corpus between the years 2017 and 2019. Bigram surprisal was calculated as the negative log of the frequency of a given two-word phrase in the Google corpus divided by the frequency of the first word of the phrase (see Appendix 2 for details). All conditions were matched for all linguistic variables across the first sentence, second sentence, and the entire vignette.

## Procedure

We used a 'magic detection' task to encourage participants to process the meaning of the vignettes without making explicit causality judgments. Participants saw 'magical' catch trials that closely resembled the experimental trials but were fantastical (e.g. 'Sadie forgot to wash her face after she ran in the heat. Now she has a cucumber nose.'). On each trial, participants indicated via button press whether 'something magical' occurred in the vignette (Yes/No). This semantic foil detection task encouraged participants to attend to the meaning of the critical vignettes while reading as naturally as possible. We required participants to press a button on every trial to ensure they were attending to the stimuli. Both sentences in a given vignette were presented simultaneously for 7 s, one above the other, followed by a 12 s inter-trial interval. Each participant saw 38 trials per condition (152 trials) plus 36 'magical' catch trials (188 total trials) in one of two versions, counterbalanced across participants, such that individual participants did not see the same sentence in both causal and noncausal vignettes. The two stimulus versions had similar meanings but different surface forms (e.g. 'Luna stood by coughing travelers on the train…' vs. 'Hugh sat by sneezing passengers on the subway…').

The experiment was divided into six 10 min runs containing six to seven trials per condition per run presented in a pseudorandom order. Vignettes from the same experimental condition repeated no more than twice consecutively, vignettes that shared similar phrase structure never repeated within a run, vignettes that referred to the same illness never repeated consecutively, and vignettes from each condition, including catch trials, were equally distributed in time across the course of the experiment.

## Mentalizing localizer experiment

To test the relationship between neural responses to inferences about the body and the mind, and to localize animacy regions, we used a localizer task to identify the mentalizing network in each participant (*Saxe and Kanwisher, 2003*; *Dodell-Feder et al., 2011*; https://saxelab.mit.edu/use-our-efficient-false-belief-localizer/). In this task, participants read 10 mentalizing stories (e.g. a protagonist has a false belief about an object's location) and 10 physical stories (physical representations depicting outdated scenes, e.g., a photograph showing an object that has since been removed) before answering a true/false comprehension question. We used the mentalizing stories from the original localizer but created new stimuli for the physical stories condition. Our physical stories incorporated more vivid descriptions of physical interactions and did not make any references to human agents, enabling us to use the mentalizing localizer as a localizer for animacy. The new physical stories were also linguistically matched to the mentalizing stories to reduce linguistic confounds (see *Shain et al., 2023*). Specifically, we matched physical and mentalizing stories (pairwise t-tests, all ps >0.3, no statistical correction) for number of characters, number of words, average number of characters per word, average syntactic dependency length, average word frequency, and average bigram surprisal, as was done for the causal inference vignettes. A comparison of both localizer versions in three pilot participants can be found in *Figure 1—figure supplement 14*.

Trials were presented in an event-related design, with each one lasting 16 s (12 s stories + 4 s comprehension question) followed by a 12 s inter-trial interval. Participants completed 2 5 min runs of the task, with trial order counterbalanced across runs and participants. The mentalizing network was identified in individual participants by contrasting *mentalizing stories > physical stories* (*Saxe and Kanwisher, 2003*; *Dodell-Feder et al., 2011*).

## Language/logic localizer experiment

To test for the presence of domain-general responses to causal inference in the language and logic networks (e.g. *Kuperberg et al., 2006*; *Operskalski and Barbey, 2017*), we used an additional localizer task. The task had three conditions: language, logic, and math. In the language condition, participants judged whether two visually presented sentences, one in active and one in passive voice, shared the same meaning. In the logic condition, participants judged whether two logical statements were consistent (e.g. *If either not Z or not Y then X* vs. *If not X then both Z and Y*). In the math condition, participants judged whether the variable *X* had the same value across two equations (for details, see *Liu et al., 2020*). Trials lasted 20 s (1 s fixation + 19 s display of stimuli) and were presented in an event-related design. Participants completed two 9 min runs of the task, with trial order counterbalanced across runs and participants. Following prior studies, the language network was identified in individual participants by contrasting *language > math* and the logic network by contrasting *logic > language* (*Monti et al., 2009*; *Kanjlia et al., 2016*; *Liu et al., 2020*).

## Data acquisition

Whole-brain fMRI data was acquired at the F.M. Kirby Research Center for Functional Brain Imaging on a 3T Phillips Achieva Multix X-Series scanner. T1-weighted structural images were collected in 150 axial slices with 1 mm isotropic voxels using the magnetization-prepared rapid gradient-echo (MP-RAGE) sequence. Functional T2*-weighted BOLD scans were collected using a gradient echo planar imaging (EPI) sequence with the following parameters: 36 sequential ascending axial slices, repetition time (TR)=2 s, echo time (TE)=0.03 s, flip angle = 70°, field of view (FOV) matrix = 76 × 70, slice thickness = 2.5 mm, inter-slice gap = 0.5, slice-coverage FH = 107.5, voxel size = 2.4×2.4×3 mm$^3$, PE direction = L/R, first order shimming. Data were acquired in one experimental session lasting approximately 120 min. All stimuli were visually presented on a rear projection screen with a Cambridge Research Systems BOLDscreen 32 UHD LCD display (image resolution = 1920 × 1080) using custom scripts written in PsychoPy3 (https://www.psychopy.org/, *Peirce et al., 2019*). Participants viewed the screen via a front-silvered, 45° inclined mirror attached to the top of the head coil.

## fMRI data preprocessing and GLM analysis

Preprocessing included motion correction, high-pass filtering (128 s), mapping to the cortical surface (Freesurfer), spatially smoothing on the surface (6 mm FWHM Gaussian kernel), and prewhitening to remove temporal autocorrelation. Covariates of no interest included signal from white matter, cerebral spinal fluid, and motion spikes.

For the main causal inference experiment, the generalized linear model (GLM) modeled the four main conditions (*Illness-Causal, Mechanical-Causal, Noncausal-Illness First, Noncausal-Mechanical First*) and the 'magical' catch trials during the 7 s display of the vignettes after convolving with a canonical hemodynamic response function and its first temporal derivative. The GLM additionally included participant response time and number of people in each vignette as covariates of no interest. For the mentalizing localizer experiment, a separate predictor was included for each condition (*mentalizing stories, physical stories*), modeling the 16 s display of each story and corresponding comprehension question. For the language/logic localizer experiment, a separate predictor was included for each of the three conditions (*language, logic, math*), modeling the 20 s duration of each trial.

For each task, runs were modeled separately and combined within-subject using a fixed-effects model (*Dale et al., 1999*; *Smith et al., 2004*). Group-level random-effects analyses were corrected for multiple comparisons across the whole cortex at p<0.05 family-wise error rate (FWER) using a nonparametric permutation test (cluster-forming threshold p<0.01 uncorrected) (*Winkler et al., 2014*; *Eklund et al., 2016*; *Eklund et al., 2019*).

## Individual-subject fROI analysis: univariate

We defined individual-subject fROIs in the PC and TPJ, as well as in the language (frontal and temporal search spaces) and logic networks. In an exploratory analysis, we defined individual-subject fROIs in an anterior parahippocampal region (i.e. anterior PPA). For all analyses, percent signal change (PSC) was extracted and averaged over the entire duration of the trial (17 s total), starting at 4 s to account for hemodynamic lag.

Illness inference fROIs were created in bilateral PC and TPJ group search spaces (*Dufour et al., 2013*) using an iterated leave-one-run-out procedure, which allowed us to perform sensitive individual-subjects analysis while avoiding statistical nonindependence (*Vul and Kanwisher, 2011*). In each participant, we identified the most illness inference-responsive vertices in bilateral PC and TPJ search spaces separately in five of the six runs (top 5% of vertices, *Illness-Causal>Mechanical-Causal*). We then extracted PSC for each condition compared to rest in the held-out run (*Illness-Causal*, *Mechanical-Causal*, *Noncausal-Illness First*, *Noncausal-Mechanical First*), averaging the results across all iterations. We used the same approach to create mechanical inference fROIs in bilateral anterior PPA search spaces from a previous study on place word representations (*Hauptman et al., 2025*). All aspects of this analysis were the same as those described above, except that the most mechanical inference-responsive vertices (top 5%, *Mechanical-Causal>Illness-Causal*) were selected.

Mentalizing fROIs were created by selecting the most mentalizing-responsive vertices (top 5%) in bilateral PC and TPJ search spaces (*Dufour et al., 2013*) using the *mentalizing stories>physical stories* contrast from the mentalizing localizer. Language fROIs were identified by selecting the most language-responsive vertices (top 5%) in left frontal and temporal language areas (search spaces: *Fedorenko et al., 2010*) using the *language>math* contrast from the language/logic localizer. A logic-responsive fROI was identified by selecting the most logic-responsive vertices (top 5%) in a left fronto-parietal network (search space: *Liu et al., 2020*) using the *logic>language* contrast. In each fROI, we extracted PSC for all conditions in the causal inference experiment.

## Individual-subject fROI analysis: multivariate

We performed MVPA (PyMVPA toolbox; *Hanke et al., 2009*) to test whether patterns of activity in the PC, TPJ, language network, and logic network distinguished illness inferences from mechanical inferences. In each participant, we identified the top 300 vertices most responsive to the mentalizing localizer (*mentalizing stories>physical stories*) in bilateral PC and TPJ search spaces (*Dufour et al., 2013*). We also identified the top 300 vertices most responsive to language (*language>math*) in a left language network search space (*Fedorenko et al., 2010*) and the top 300 vertices most responsive to logical reasoning (*logic>language*) in a left logic network search space (*Liu et al., 2020*).

In an exploratory analysis, we performed MVPA to test whether patterns of activity in the left PC and in the language and logic networks distinguished causal from noncausal vignettes. To avoid statistical nonindependence, we defined additional fROIs in the left PC for the purposes of this analysis. In each participant, we identified the top 300 vertices most responsive to the critical conditions over rest (*Illness-Causal+Mechanical-Causal+Noncausal-Illness First +Noncausal-Mechanical First>Rest*) in a left PC search space (*Dufour et al., 2013*).

For each vertex in each participant's fROIs, we obtained one observation per condition per run (z-scored beta parameter estimate of the GLM). A linear support vector machine (SVM) was then trained on data all but one of the runs and tested on the left-out run in a cross-validation procedure. Classification accuracy was averaged across all permutations of the training/test splits. We compared classifier performance within each fROI to chance (50%; one-tailed test). Significance was evaluated against an empirically generated null distribution using a combined permutation and bootstrap approach (*Schreiber and Krekelberg, 2013*; *Stelzer et al., 2013*). In this approach, t-statistics obtained for the observed data are compared against an empirically generated null distribution. We report the t-values obtained for the observed data and the nonparametric p-values, where p corresponds to the proportion of the shuffled analyses that generated a comparable or higher t-value.

The null distribution was generated using a balanced block permutation test by shuffling condition labels within run 1000 times for each subject (*Schreiber and Krekelberg, 2013*). Then, a bootstrapping procedure was used to generate an empirical null distribution for each statistical test across participants by sampling one permuted accuracy value from each participant's null distribution 15,000 times (with replacement) and running each statistical test on these permuted samples, thus generating a null distribution of 15,000 statistical values for each test (*Stelzer et al., 2013*).

## Searchlight MVPA

We used a linear SVM classifier to test decoding between all pairs of causal and noncausal conditions (i.e. *Illness-Causal* vs. *Mechanical-Causal*, *Illness-Causal* vs. *Noncausal-Mechanical First*, *Illness-Causal* vs. *Noncausal-Illness First*, *Mechanical-Causal* vs. *Noncausal-Mechanical First*, and *Mechanical-Causal*

vs. *Noncausal-Illness First*) across the whole cortex using a 10 mm radius spherical searchlight (according to geodesic distance, to better respect cortical anatomy over Euclidean distance; *Glasser et al., 2013*). This yielded for each participant five classification maps, indicating the classifier's accuracy in a neighborhood surrounding every vertex. Individual subject searchlight accuracy maps were then averaged within analysis, and the resulting group-wise maps were thresholded using the PyMVPA implementation of the two-step cluster-thresholding procedure described in *Stelzer et al., 2013* (*Hanke et al., 2009*). This procedure permutes block labels within participant to generate a null distribution within subject (100 times) and then samples from these (10,000) to generate a group-wise null distribution (as in the fROI analysis). The whole-brain searchlight maps are then thresholded using a combination of vertex-wise threshold (p<0.001 uncorrected) and cluster size threshold (FWER p<0.05, corrected for multiple comparisons across the entire cortical surface).

## Data availability statement

Custom lab software for fMRI analysis is available via GitHub (https://github.com/NPDL/NPDL-scripts copy archived at *Lane et al., 2025*). Stimuli and code specific to this project are accessible via OSF (https://osf.io/cx9n2/). fMRI and behavioral data are accessible via OpenICPSR (10.3886/E237324V1).

## Acknowledgements

We thank the F.M. Kirby Research Center for Functional Brain Imaging at the Kennedy Krieger Institute for their assistance with data collection, and the participants for making this research possible. This work was supported by a grant from the National Science Foundation (BCS-2318685 to MB).

## Additional information

### Funding

| Funder | Grant reference number | Author |
| --- | --- | --- |
| National Science Foundation | BCS-2318685 | Marina Bedny |

The funders had no role in study design, data collection and interpretation, or the decision to submit the work for publication.

### Author contributions

Miriam Hauptman, Conceptualization, Data curation, Software, Formal analysis, Investigation, Visualization, Methodology, Writing – original draft, Writing – review and editing; Marina Bedny, Conceptualization, Resources, Supervision, Funding acquisition, Methodology, Writing – review and editing

### Author ORCIDs

Miriam Hauptman ⓘ https://orcid.org/0000-0002-5903-1552

### Ethics

All participants gave written informed consent. The study was reviewed and approved by the Johns Hopkins Medicine Institutional Review Boards.

Reviewer #1 (Public review): https://doi.org/10.7554/eLife.101944.3.sa1
Reviewer #2 (Public review): https://doi.org/10.7554/eLife.101944.3.sa2
Reviewer #3 (Public review): https://doi.org/10.7554/eLife.101944.3.sa3
Author response https://doi.org/10.7554/eLife.101944.3.sa4

## Additional files

### Supplementary files

Supplementary file 1. Illness types present in the stimulus set.

Supplementary file 2. Results of preregistered MVPA for *Illness-Causal* vs. *Mechanical-Causal* in individual-subject functional ROIs (fROI). Each fROI was created by selecting the top 300 vertices for each contrast (see 'Contrast') in each search space. Accuracy refers to classifier performance against chance (50%) for *Illness-Causal* vs. *Mechanical-Causal*. Permuted and Bonferroni-corrected (across fROIs) p-values are reported. Ment_vs_phys: *mentalizing stories>physical stories* (mentalizing localizer). Caus_vs_rest: *Illness-Causal+Illness-Mechanical>Rest*. Logic_vs_lang: *logic >language* (language/logic localizer). Lang_vs_math: *language>math* (language/logic localizer). Visualizations of these results are displayed in *Figure 1—figure supplement 11*.

Supplementary file 3. MVPA results for all tests in select individual-subject functional ROIs (fROI). Each fROI was created by selecting the top 300 vertices for each contrast in each search space: left PC (LPC)=top *main experimental conditions>rest,* language = top *language>math* (language/logic localizer), logic = top *logic >language* (language/logic localizer). Accuracy refers to classifier performance against chance (50%) for each test. Permuted and Bonferroni-corrected (across fROIs) p-values are reported. Visualizations of these results are displayed in *Figure 1—figure supplement 12*.

MDAR checklist

## Data availability

fMRI and behavioral data are publicly available via OpenICPSR (https://doi.org/10.3886/E237324V1).

The following dataset was generated:

| Author(s) | Year | Dataset title | Dataset URL | Database and Identifier |
|-----------|------|---------------|-------------|-------------------------|
| Hauptman M, Bedny M | 2025 | The neural basis of causal inferences about biological and physical processes | https://doi.org/10.3886/E237324V1 | OpenICPSR, 10.3886/E237324V1 |

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

## Appendix 1

### Online experiment protocol

Prior to the fMRI experiment, we collected explicit causality judgments from a separate group of online participants (*n*=30). Each online participant read all vignettes from the causal inference experiment (152 vignettes) in addition to 12 filler vignettes that were designed to be either maximally causally related or unrelated (164 vignettes total), one vignette at a time. Their task was to judge the extent to which it was possible that the event described in the first sentence of each vignette caused the event described in the second sentence on a four-point scale (1=not possible; 4=very possible). Four participants were excluded on the basis of inaccurate responses on the filler trials (i.e. difference between average ratings for maximally causally related and maximally causally unrelated vignettes <2). Among the 26 remaining participants, 12 read vignettes from Version A and 14 read vignettes from Version B of the experiment. To eliminate erroneous responses, we first excluded trials with RTs 2.5 SD outside their respective condition means within participants and then excluded trials with outlier RTs (more than 1.5 IQR below Q1 or more than 1.5 IQR above Q3) across participants (approximately 5% of all trials excluded in total). We found that both causal conditions (*Illness-Causal*, *Mechanical-Causal*) were more causally connected than both noncausal conditions, $t(25) = 36.97$, $p<0.001$ (causal: *M*=3.51 ± 0.78 SD, noncausal: *M*=1.10 ± 0.45 SD). In addition, *Illness-Causal* and *Mechanical-Causal* items received equally high causality ratings, $t(25) = -0.64$, $p=0.53$ (*Illness-Causal: M*=3.49 ± 0.77 SD, *Mechanical-Causal: M*=3.53 ± 0.79 SD).

## Appendix 2

### Details on measuring linguistic variables

All conditions were matched (pairwise t-tests, all ps>0.3, no statistical correction) on multiple linguistic variables known to modulate neural activity in language regions (e.g. *Pallier et al., 2011*; *Shain et al., 2020*). These included number of characters, number of words, average number of characters per word, average word frequency, average bigram surprisal (Google Books Ngram Viewer, https://books.google.com/ngrams/), and average syntactic dependency length (Stanford Parser; *Marneffe et al., 2006*). Sentences that were incorrectly parsed by the automatic syntactic parser (i.e. past participle adjectives parsed as verbs) were corrected by hand. Word frequency was calculated as the negative log of a word's occurrence rate in the Google corpus between the years 2017 the 2019. Bigram surprisal was calculated as the negative log of the frequency of a given two-word phrase in the Google corpus divided by the frequency of the first word of the phrase.

This calculation uses a log base of 2 in order to express surprisal in terms of 'bits' that the first word provides in the context of the phrase. We used *bigram* surprisal as our surprisal measure to maximize the number of n-grams that had an entry in the corpus. Even so, 64 out of the 1515 total bigrams (4%) did not have an entry in the corpus and were therefore assigned the highest surprisal value among the rest of the bigrams (see *Willems et al., 2016*).

## Appendix 3

### Full behavioral results

Accuracy on the magic detection task was at ceiling (*M*=97.9% ± 2.2 SD). There were no significant differences across the four main experimental conditions (*Illness-Causal*, *Mechanical-Causal*, *Noncausal-Illness First, Noncausal-Mechanical First*), but participants were more accurate on *Illness-Causal* trials compared to 'magical' catch trials ($F_{(4,76)}$ = 2.81, p=0.03; *Illness-Causal:* M=98.8% ± 2.2 SD; 'magical' catch trials*: M*=96.4% ± 3.8 SD).

A one-way repeated-measures ANOVA evaluating response time revealed a main effect of condition, $F_{(4,76)}$ = 8.17, p<0.001, whereby participants were faster on *Illness-Causal* trials (*M*=4.73 ± 0.81 SD) compared to *Noncausal-Illness First* (*M*=5.33 s±0.85 SD), *Noncausal-Mechanical First* (*M*=5.27 s±0.89 SD) trials, and 'magical' catch trials (*M*=5.34 s±0.89 SD). There were no differences in response time between *Mechanical-Causal* (*M*=5.15 s±0.88 SD) and any other conditions.

Accuracy on the language/logic localizer task was significantly lower for the logic task compared to both the language and math tasks (logic: *M*=67.5% ± 14.0 SD, math: *M*=93.8% ± 6.4 SD, language: *M*=98.1% ± 5.8 SD; $F_{(2,38)}$ = 60.38, p<0.0001). Similarly, response time was slowest on the logic task, followed by math and then language (logic: *M*=8.78 s±1.88 SD, math: *M*=6.20 s±1.37 SD, language: *M*=5.18 s±1.53 SD; $F_{(2,38)}$ = 44.28, p<0.001).

Accuracy on the mentalizing localizer task was not different across the mentalizing stories and physical stories conditions (mentalizing: 83.50%±15.7 SD, physical: 90.50%±12.3 SD; $F_{(1,19)}$ = 2.73, p=0.12). However, response time for the mentalizing stories was significantly slower (mentalizing: 3.46 s±0.55 SD, physical: 3.11 s±0.56 SD; $F_{(1,19)}$ = 16.59, p<0.001).

## Appendix 4

### Individual-subject univariate fROI analysis in the (FFA)

In an exploratory analysis, we defined individual-subject fROIs in the FFA.

Illness inference fROIs were created in left and right FFA search spaces from a previous study on responses to images of faces in the ventral stream (*Julian et al., 2012*) using an iterated leave-one-run-out procedure. In each participant, we identified the most illness inference-responsive vertices in left and right FFA search spaces in five of the six runs (top 5% of vertices, *Illness-Causal>Mechanical-Causal*). We then extracted PSC for each condition compared to rest in the held-out run (*Illness-Causal*, *Mechanical-Causal*, *Noncausal-Illness First, Noncausal-Mechanical First*), averaging the results across all iterations.

In contrast to the PC, the FFA did not show a preference for illness inferences compared to mechanical inferences (leave-one-run-out individual-subject fROI analysis; repeated-measures ANOVA, condition (*Illness-Causal*, *Mechanical-Causal*) × hemisphere (left, right): main effect of condition, $F_{(1,19)}$ = 0.04, p=0.84, main effect of hemisphere, $F_{(1,19)}$ = 9.46, p=0.006, condition × hemisphere interaction, $F_{(1,19)}$ = 1.34, p=0.26; *Figure 1—figure supplement 13*). Additionally, the FFA did not show a preference for illness inferences compared to noncausal vignettes, which contained illness-related language but were not causally connected (repeated-measures ANOVA, condition (*Illness-Causal*, *Noncausal-Illness First*) × hemisphere (left, right): main effect of condition, $F_{(1,19)}$ = 0.94, p=0.34, main effect of hemisphere, $F_{(1,19)}$ = 4.47, p=0.05, condition × hemisphere interaction, $F_{(1,19)}$ = 0.06, p=0.82; repeated-measures ANOVA, condition (*Illness-Causal*, *Noncausal-Mechanical First*) × hemisphere (left, right): main effect of condition, $F_{(1,19)}$ = 0.07, p=0.8; main effect of hemisphere, $F_{(1,19)}$ = 7.59, p=0.01; condition × hemisphere interaction, $F_{(1,19)}$ = 2.72, p=0.12; *Figure 1—figure supplement 13*). Thus, although the FFA exhibits a preference for images of animates (e.g. *Kanwisher et al., 1997*), the current evidence suggests that this region is not sensitive to abstract causal knowledge about animacy-specific processes (i.e. illness).

