## [Editor Report · eLife Assessment]

This study investigates the neural basis of causal inference of illness, suggesting that it relies on semantic networks specific to living things in the absence of a generalized representation of causal inference across domains. The main hypothesis is **compelling**, and is supported by **solid** methods and data analysis. Overall, the findings make a **valuable** contribution to understanding the role of domain-specific semantic networks, particularly the precuneus, in implicit causal inference about illness.

---

## [Referee Report · Reviewer #1 (Public review)]

Summary:

In this study, the authors aim to understand the neural basis of implicit causal inference, specifically how people infer causes of illness. They use fMRI to explore whether these inferences rely on content-specific semantic networks or broader, domain-general neurocognitive mechanisms. The study explores two key hypotheses: first, that causal inferences about illness rely on semantic networks specific to living things, such as the 'animacy network,' given that illnesses affect only animate beings; and second, that there might be a common brain network supporting causal inferences across various domains, including illness, mental states, and mechanical failures. By examining these hypotheses, the authors aim to determine whether causal inferences are supported by specialized or generalized neural systems.

The authors observed that inferring illness causes selectively engaged a portion of the precuneus (PC) associated with the semantic representation of animate entities, such as people and animals. They found no cortical areas that responded to causal inferences across different domains, including illness and mechanical failures. Based on these findings, the authors concluded that implicit causal inferences are supported by content-specific semantic networks, rather than a domain-general neural system, indicating that the neural basis of causal inference is closely tied to the semantic representation of the specific content involved.

Strengths:

- The inclusion of the four conditions in the design is well thought out, allowing for the examination of the unique contribution of causal inference of illness compared to either a different type of causal inference (mechanical) or non-causal conditions. This design also has the potential to identify regions involved in a shared representation of inference across general domains.

- The presence of the three localizers for language, logic, and mentalizing, along with the selection of specific regions of interest (ROIs), such as the precuneus and anterior ventral occipitotemporal cortex (antVOTC), is a strong feature that supports a hypothesis-driven approach (although see below for a critical point related to the ROI selection).

- The univariate analysis pipeline is solid and well developed.

- The statistical analyses are a particularly strong aspect of the paper.

Weaknesses:

After carefully considering the authors' response, I believe that my primary concern has not been fully addressed. My main point remains unresolved:

The authors attempt to test for the presence of a shared network by performing only the Causal vs. Non-causal analysis. However, this approach is not sufficiently informative because it includes all conditions mixed together and does not clarify whether both the illness-causal and mechanical-causal conditions contribute to the observed results.

To address this limitation, I originally suggested an additional step: using as ROIs the different regions that emerged in the Causal vs. Non-causal decoding analysis and conducting four separate decoding analyses within these specific clusters:

(1) Illness-Causal vs. Non-causal - Illness First

(2) Illness-Causal vs. Non-causal - Mechanical First

(3) Mechanical-Causal vs. Non-causal - Illness First

(4) Mechanical-Causal vs. Non-causal - Mechanical First

This approach would allow the authors to determine whether any of these ROIs can decode both the illness-causal and mechanical-causal conditions against at least one non-causal condition. However, the authors did not conduct these analyses, citing an independence issue. I disagree with this reasoning because these analyses would serve to clarify their initial general analysis, in which multiple conditions were mixed together. As the results currently stand, it remains unclear which specific condition is driving the effects.

My suggestion was to select the ROIs from their general analysis (Causal vs. Non-causal) and then examine in more detail which conditions were driving these results. This is not a case of double-dipping from my perspective, but rather a necessary step to unpack the general findings. Moreover, using ROIs would actually reduce the number of multiple comparisons that need to be controlled for.

If the authors believe that this approach is methodologically incorrect, then they should instead conduct all possible analyses at the whole-brain level to examine the effects of the specific conditions independently.

---

## [Referee Report · Reviewer #2 (Public review)]

Summary:

In this study, the authors test whether intuitive biological causal knowledge is embedded in domain-specific semantic networks, primarily focusing on the precuneus as part of the animacy semantic network. They do so tanks to an fMRI task, by comparing brain activity elicited by participants' exposure to written situations suggesting a plausible cause of illness with brain activity in linguistically equivalent situations suggesting a plausible cause of mechanical failure or damage and non-causal situations. These contrasts confirm the PC as the main "culprit" in whole-brain and fROIs univariate analyses. In turn, inferring causes of mechanical failure engages mostly the PPA. The authors further test whether the content-specificity has to do with inferences about animates in general, or if there are some distinctions between reasoning about people's bodies versus mental states. To answer this question, the authors localize the mentalizing network and study the relation between brain activity elicited by Illness-Causal > Mech-Causal and Mentalizing > Physical stories. They conclude that inferring about the causes of illness partially differentiates from reasoning about people's states of mind. The authors finally test the alternative yet non-mutually exclusive hypothesis that both types of implicit causal inferences (illness and mechanical) depend on shared neural machinery. Good candidates are language and logic, which justifies the use of a language/logic localizer. No evidence of commonalities across causal inferences versus non-causal situations are found.

Strengths:

(1) This study introduces a useful paradigm and well-designed set of stimuli to test for implicit causal inferences.

(2) Another important methodological advance is the addition of physical stories to the original mentalizing protocol.

These tools pave the way for further investigation of domain-specific causal inference.

(3) The authors have significantly improved the manuscript, addressing previous concerns and incorporating additional analyses that strengthen their conclusions.

Key improvements:

(1) The revised introduction makes the study's contribution more explicit and resolves initial ambiguities regarding its scope.

(2) The rationale for focusing primarily on the precuneus is now clearer and the additional analysis in the fusiform face area provides a valuable comparison.

(3) The revised manuscript now includes a more detailed examination of the searchlight MVPA results, showing that illness and mechanical inferences elicit spatially distinct neural patterns in key regions, including the left PC, anterior PPA, and lateral occipitotemporal cortex.

(4) The authors' justification for using an implicit inference task, arguing that explicit tasks introduce executive function confounds, is convincing.

(5) The authors now acknowledge that while their results support a content-specific neural basis for implicit causal inference, domain-general mechanisms may still play a role in other contexts.

I have no major remaining concerns.

---

## [Referee Report · Reviewer #3 (Public review)]

Summary:

This study employed an implicit task, showing vignettes to participants while bold signal was acquired. The aim was to capture automatic causal inferences that emerge during language processing and comprehension. In particular, the authors compared causal inferences about illness with two control conditions, causal inferences about mechanical failures and non-causal phrases related to illnesses. All phrases that where employed described contexts with people, to avoid animacy/inanimate confound in the results. The authors had a specific hypothesis concerning the role of the precuneus (PC) being sensitive to causal inferences about illnesses (that was preregistered).

Findings indicate that implicit causal inferences are facilitated by semantic networks specialized for encoding causal knowledge.

Strengths:

The major strength of the study is the clever design of the stimuli (which are nicely matched for a number of features) which can tease apart the role of the type of causal inference (illness-causal or mechanical-causal) and the use of two localizers (logic/language and mentalizing) to investigate the hypothesis that the language and/or logical reasoning networks preferentially respond to causal inference regardless of the content domain being tested (illnesses or mechanical).

I think that authors' revisions of the original manuscript have strengthened the study. Overall, the paper provides an interesting contribution to the (rather new) field of study concerning the neural basis of implicit causal inference.

I see two weaknesses concerning the visualization of the data (which could be improved)

(1) Measures of dispersion are now provided for the average PSC in the critical window. It would be more appropriate to show the variance of the data also for the percentage signal changes (PSC) figures (e.g., 1A by using shaded lines providing SE around the means or boxplots at each timepoint).

(2) The authors could consider showing in Figure 2 the data of supplementary Figure 3. It is not clear why the authors report in the main manuscript the results of a subsample of participants (and only for this figure).

---

## [Author Response]

The following is the authors’ response to the original reviews

**Reviewer #1 (Public review):**
Summary:In this study, the authors aim to understand the neural basis of implicit causal inference, specifically how people infer causes of illness. They use fMRI to explore whether these inferences rely on content-specific semantic networks or broader, domain-general neurocognitive mechanisms. The study explores two key hypotheses: first, that causal inferences about illness rely on semantic networks specific to living things, such as the 'animacy network,' given that illnesses affect only animate beings; and second, that there might be a common brain network supporting causal inferences across various domains, including illness, mental states, and mechanical failures. By examining these hypotheses, the authors aim to determine whether causal inferences are supported by specialized or generalized neural systems.The authors observed that inferring illness causes selectively engaged a portion of the precuneus (PC) associated with the semantic representation of animate entities, such as people and animals. They found no cortical areas that responded to causal inferences across different domains, including illness and mechanical failures. Based on these findings, the authors concluded that implicit causal inferences are supported by content-specific semantic networks, rather than a domain-general neural system, indicating that the neural basis of causal inference is closely tied to the semantic representation of the specific content involved.Strengths:(1) The inclusion of the four conditions in the design is well thought out, allowing for the examination of the unique contribution of causal inference of illness compared to either a different type of causal inference (mechanical) or non-causal conditions. This design also has the potential to identify regions involved in a shared representation of inference across general domains.(2) The presence of the three localizers for language, logic, and mentalizing, along with the selection of specific regions of interest (ROIs), such as the precuneus and anterior ventral occipitotemporal cortex (antVOTC), is a strong feature that supports a hypothesis-driven approach (although see below for a critical point related to the ROI selection).(3) The univariate analysis pipeline is solid and well-developed.(4) The statistical analyses are a particularly strong aspect of the paper.Weaknesses:Based on the current analyses, it is not yet possible to rule out the hypothesis that inferring illness causes relies on neurocognitive mechanisms that support causal inferences irrespective of their content, neither in the precuneus nor in other parts of the brain.(1) The authors, particularly in the multivariate analyses, do not thoroughly examine the similarity between the two conditions (illness-causal and mechanical-causal), as they are more focused on highlighting the differences between them. For instance, in the searchlight MVPA analysis, an interesting decoding analysis is conducted to identify brain regions that represent illness-causal and mechanical-causal conditions differently, yielding results consistent with the univariate analyses. However, to test for the presence of a shared network, the authors only perform the Causal vs. Non-causal analysis. This analysis is not very informative because it includes all conditions mixed together and does not clarify whether both the illness-causal and mechanical-causal conditions contribute to these results.(2) To address this limitation, a useful additional step would be to use as ROIs the different regions that emerged in the Causal vs. Non-causal decoding analysis and to conduct four separate decoding analyses within these specific clusters:(a) Illness-Causal vs. Non-causal - Illness First;(b) Illness-Causal vs. Non-causal - Mechanical First;(c) Mechanical-Causal vs. Non-causal - Illness First;(d) Mechanical-Causal vs. Non-causal - Mechanical First.This approach would allow the authors to determine whether any of these ROIs can decode both the illness-causal and mechanical-causal conditions against at least one non-causal condition.(3) Another possible analysis to investigate the existence of a shared network would be to run the searchlight analysis for the mechanical-causal condition versus the two non-causal conditions, as was done for the illness-causal versus non-causal conditions, and then examine the conjunction between the two. Specifically, the goal would be to identify ROIs that show significant decoding accuracy in both analyses.

The hypothesis that a neural mechanism supports causal inference across domains predicts higher univariate responses when causal inferences occur than when they do not. This prediction was not generated by us ad hoc but rather has been made by almost all previous cognitive neuroscience papers on this topic (Ferstl & von Cramon, 2001; Satpute et al., 2005; Fugelsang & Dunbar, 2005; Kuperberg et al., 2006; Fenker et al., 2010; Kranjec et al., 2012; Pramod, Chomik-Morales, et al., 2023; Chow et al., 2008; Mason & Just, 2011; Prat et al., 2011). Contrary to this hypothesis, we find that the precuneus (PC) is most activated for illness inferences and most deactivated for mechanical inferences relative to rest, suggesting that the PC does not support domain-general causal inference. To further probe the selectivity of the PC for illness inferences, we created group overlap maps that compare PC responses to illness inferences and mechanical inferences across participants. The PC shows a strong preference for illness inferences and is therefore unlikely to support causal inferences irrespective of their content (Supplementary Figures 6 and 7). We also note that, in whole-cortex analysis, no shared regions responded more to causal inference than noncausal vignettes across domains. Therefore, the prediction made by the ‘domain-general causal engine’ proposal as it has been articulated in the literature is not supported in our data.

Taking a multivariate approach, the hypothesis that a neural mechanism supports causal inference across domains also predicts that relevant regions can decode between all possible pairs of causal vs. noncausal conditions (e.g., Illness-Causal vs. Noncausal-Illness First, Mechanical-Causal vs. Noncausal-Illness First, etc.). The analysis described by the reviewer in (2), in which the regions that distinguish between causal vs. noncausal conditions in searchlight MVPA are used as ROIs to test various causal vs. noncausal contrasts, is non-independent. Therefore, we cannot perform this analysis. In accordance with the reviewer’s suggestions in (3), now include searchlight MVPA results for the mechanical inference condition compared to the two noncausal conditions (Supplementary Figure 9). No regions are shared across the searchlight analyses comparing all possible pairs of causal and noncausal conditions, providing further evidence that there are no shared neural responses to causal inference in our dataset.

(4) Along the same lines, for the ROI MVPA analysis, it would be useful not only to include the illness-causal vs. mechanical-causal decoding but also to examine the illness-causal vs. non-causal conditions and the mechanical-causal vs. non-causal conditions. Additionally, it would be beneficial to report these data not just in a table (where only the mean accuracy is shown) but also using dot plots, allowing the readers to see not only the mean values but also the accuracy for each individual subject.

We have performed these analyses and now include a table of the results as well as figures displaying the dispersion across participants (Supplementary Tables 2 and 3, Supplementary Figures 10 and 11). In the left PC, the illness inference condition was decoded from one of the noncausal conditions, and the mechanical inference condition was decoded from the same noncausal condition. The language network did not decode between any causal/noncausal pairs. In the logic network, the illness inference condition was decoded from one of the noncausal conditions, and the mechanical inference condition was decoded from the other noncausal condition. Thus, no regions showed the predicted ‘domain-general’ pattern, i.e., significant decoding between all causal/noncausal pairs.

Importantly, the decoding results must be interpreted in light of significant univariate differences across conditions (e.g., greater responses to illness inferences compared to noncausal vignettes in the PC). Linear classifiers are highly sensitive to univariate differences (Coutanche, 2013; Kragel et al., 2012; Hebart & Baker, 2018; Woolgar et al., 2014; Davis et al., 2014; Pakravan et al., 2022).

(5) The selection of Regions of Interest (ROIs) is not entirely straightforward:In the introduction, the authors mention that recent literature identifies the precuneus (PC) as a region that responds preferentially to images and words related to living things across various tasks. While this may be accurate, we can all agree that other regions within the ventral occipital-temporal cortex also exhibit such preferences, particularly areas like the fusiform face area, the occipital face area, and the extrastriate body area. I believe that at least some parts of this network (e.g., the fusiform gyrus) should be included as ROIs in this study. This inclusion would make sense, especially because a complementary portion of the ventral stream known to prefer non-living items (i.e., anterior medial VOTC) has been selected as a control ROI to process information about the mechanical-causal condition. Given the main hypothesis of the study - that causal inferences about illness might depend on content-specific semantic representations in the 'animacy network' - it would be worthwhile to investigate these ROIs alongside the precuneus, as they may also yield interesting results.

We thank the reviewer for their suggestion to test the FFA region. We think this provides an interesting comparison to the PC and hypothesized that, in contrast to the PC, the FFA does not encode abstract causal information about animacy-specific processes (i.e., illness). As we mention in the Introduction, although the fusiform face area (FFA) also exhibits a preference for animates, it does so primarily for images in sighted people (Kanwisher et al., 1997; Kanwisher et al., 1997; Grill-Spector et al., 2004; Noppeney et al., 2006; Konkle & Caramazza, 2013; Connolly et al., 2016; Bi et al., 2016).

We did not select the FFA as a region of interest when preregistering the current study because we did not predict it would show sensitivity to causal knowledge. In accordance with the reviewer’s suggestions, we now include the FFA as an ROI in individual-subject univariate analysis (Supplementary Figure 8, Appendix 4). Because we did not run a separate FFA localizer task when collecting the data, we used FFA search spaces from a previous study investigating responses to face images (Julian et al., 2012). We followed the same analysis procedure that was used to investigate responses to illness inferences in the PC. Neither left nor right FFA exhibited a preference for illness inferences compared to mechanical inferences or to the noncausal conditions. This result is interesting and is now briefly discussed in the Discussion section.

(6) Visual representation of results:In all the figures related to ROI analyses, only mean group values are reported (e.g., Figure 1A, Figure 3, Figure 4A, Supplementary Figure 6, Figure 7, Figure 8). To better capture the complexity of fMRI data and provide readers with a more comprehensive view of the results, it would be beneficial to include a dot plot for a specific time point in each graph. This could be a fixed time point (e.g., a certain number of seconds after stimulus presentation) or the time point showing the maximum difference between the conditions of interest. Adding this would allow for a clearer understanding of how the effect is distributed across the full sample, such as whether it is consistently present in every subject or if there is greater variability across individuals.

We thank the reviewer for this suggestion. We now include scattered box plots displaying the dispersion in average percent signal change across participants in Figures 1, 3, and 4, and Supplementary Figures 8, 12, and 14.

(7) Task selection:(a) To improve the clarity of the paper, it would be helpful to explain the rationale behind the choice of the selected task, specifically addressing: (i) why an implicit inference task was chosen instead of an explicit inference task, and (ii) why the "magic detection" task was used, as it might shift participants' attention more towards coherence, surprise, or unexpected elements rather than the inference process itself.(b) Additionally, the choice to include a large number of catch trials is unusual, especially since they are modeled as regressors of non-interest in the GLM. It would be beneficial to provide an explanation for this decision.

We chose an orthogonal foil detection task, rather than an explicit causal judgment task, to investigate automatic causal inferences during reading and to unconfound such processing as much as possible from explicit decision-making processes (see Kuperberg et al., 2006 for discussion). Analogous foil detection paradigms have been used to study sentence processing and word recognition (Pallier et al., 2011; Dehaene-Lambertz et al., 2018). We now clarify this in the Introduction. The “magical” element occurred both within and across sentences so that participants could not use coherence as a cue to complete the task. Approximately 1/5 (19%) of the trials were magical catch trials to ensure that participants remained attentive throughout the experiment.

**Reviewer #2 (Public review):**
Summary:In this study, the authors hypothesize that "causal inferences about illness depend on content-specific semantic representations in the animacy network". They test this hypothesis in an fMRI task, by comparing brain activity elicited by participants' exposure to written situations suggesting a plausible cause of illness with brain activity in linguistically equivalent situations suggesting a plausible cause of mechanical failure or damage and non-causal situations. These contrasts identify PC as the main "culprit" in a whole-brain univariate analysis. Then the question arises of whether the content-specificity has to do with inferences about animates in general, or if there are some distinctions between reasoning about people's bodies versus mental states. To answer this question, the authors localize the mentalizing network and study the relation between brain activity elicited by Illness-Causal > Mech-Causal and Mentalizing > Physical stories. They conclude that inferring about the causes of illness partially differentiates from reasoning about people's states of mind. The authors finally test the alternative yet non-mutually exclusive hypothesis that both types of causal inferences (illness and mechanical) depend on shared neural machinery. Good candidates are language and logic, which justifies the use of a language/logic localizer. No evidence of commonalities across causal inferences versus non-causal situations is found.Strengths:(1) This study introduces a useful paradigm and well-designed set of stimuli to test for implicit causal inferences.(2) Another important methodological advance is the addition of physical stories to the original mentalizing protocol.(3) With these tools, or a variant of these tools, this study has the potential to pave the way for further investigation of naïve biology and causal inference.Weaknesses:(1) This study is missing a big-picture question. It is not clear whether the authors investigate the neural correlates of causal reasoning or of naïve biology. If the former, the choice of an orthogonal task, making causal reasoning implicit, is questionable. If the latter, the choice of mechanical and physical controls can be seen as reductive and problematic.

We have modified the Introduction to clarify that the primary goal of the current study is to test the claim that semantic networks encode causal knowledge – in this case, causal intuitive theories of biology. Most conceptions of intuitive biology, intuitive psychology, and intuitive physics describe them as causal frameworks (e.g., Wellman & Gelman, 1992; Simons & Keil, 1995; Keil et al., 1999; Tenenbaum, Griffiths, & Niyogi, 2007; Gopnik & Wellman, 2012; Gerstenberg & Tenenbaum, 2017). As noted above, we chose an implicit task to investigate automatic causal inferences during reading and to unconfound such processing as much as possible from explicit decision-making processes. We are not sure what the reviewer means when they say that mechanical and physical controls are reductive. This is the standard control condition in neural and behavioral paradigms that investigate intuitive psychology and intuitive biology (e.g., Saxe & Kanwisher, 2003; Gelman & Wellman, 1991).

(2) The rationale for focusing mostly on the precuneus is not clear and this choice could almost be seen as a post-hoc hypothesis.

This study is preregistered (https://osf.io/6pnqg). The preregistration states that the precuneus is a hypothesized area of interest, so this is not a post-hoc hypothesis. Our hypothesis was informed by multiple prior studies implicating the precuneus in the semantic representation of animates (e.g., people, animals) (Fairhall & Caramazza, 2013a, 2013b; Fairhall et al., 2014; Peer et al., 2015; Wang et al., 2016; Silson et al., 2019; Rabini, Ubaldi, & Fairhall, 2021; Deen & Freiwald, 2022; Aglinskas & Fairhall, 2023; Hauptman, Elli, et al., 2025). We also conducted a pilot experiment with separate participants prior to pre-registering the study. We now clarify our rationale for focusing on the precuneus in the Introduction:

“Illness affects living things (e.g., people and animals) rather than inanimate objects (e.g., rocks, machines, houses). Thinking about living things (animates) as opposed to non-living things (inanimate objects/places) recruits partially distinct neural systems (e.g., Warrington & Shallice, 1984; Hillis & Caramazza, 1991; Caramazza & Shelton, 1998; Farah & Rabinowitz, 2003). The precuneus (PC) is part of the ‘animacy’ semantic network and responds preferentially to living things (i.e., people and animals), whether presented as images or words (Devlin et al., 2002; Fairhall & Caramazza, 2013a, 2013b; Fairhall et al., 2014; Peer et al., 2015; Wang et al., 2016; Silson et al., 2019; Rabini, Ubaldi, & Fairhall, 2021; Deen & Freiwald, 2022; Aglinskas & Fairhall, 2023; Hauptman, Elli, et al., 2025). By contrast, parts of the visual system (e.g., fusiform face area) that respond preferentially to animates do so primarily for images (Kanwisher et al., 1997; Grill-Spector et al., 2004; Noppeney et al., 2006; Mahon et al., 2009; Konkle & Caramazza, 2013; Connolly et al., 2016; see Bi et al., 2016 for a review). We hypothesized that the PC represents causal knowledge relevant to animates and tested the prediction that it would be activated during implicit causal inferences about illness, which rely on such knowledge (preregistration: https://osf.io/6pnqg).”

(3) The choice of an orthogonal 'magic detection' task has three problematic consequences in this study:(a) It differs in nature from the 'mentalizing' task that consists of evaluating a character's beliefs explicitly from the corresponding story, which complicates the study of the relation between both tasks. While the authors do not compare both tasks directly, it is unclear to what extent this intrinsic difference between implicit versus explicit judgments of people's body versus mental states could influence the results.(b) The extent to which the failure to find shared neural machinery between both types of inferences (illness and mechanical) can be attributed to the implicit character of the task is not clear.(c) The introduction of a category of non-interest that contains only 36 trials compared to 38 trials for all four categories of interest creates a design imbalance.

We disagree with the reviewer’s argument that our use of an implicit “magic detection” task is problematic. Indeed, we think it is one of the advances of the current study over prior work.

a) Prior work has shown that implicit mentalizing tasks (e.g., naturalistic movie watching) engages the theory of mind network, suggesting that the implicit/explicit nature of the task does not drive the activation of this network (Jacoby et al., 2016; Richardson et al., 2018). With these data in mind, it is unlikely that the implicit/explicit nature of the causal inference and theory of mind tasks in the present experiment can explain observed differences between them.

b) Explicit causal inferences introduce a collection of executive processes that potentially confound the results and make it difficult to know whether neural signatures are related to causal inference per se. The current study focuses on the neural basis of implicit causal inference, a type of inference that is made routinely during language comprehension. We do not claim to find neural signatures of all causal inferences, we do not think any study could claim to do so because causal inferences are a highly varied class.

c) Our findings do not exclude the possibility that content-invariant responses are elicited during explicit causality judgments. We clarify this point in the Results (e.g., “These results leave open the possibility that domain-general systems support the explicit search for causal connections”) and Discussion (e.g., “The discovery of novel causal relationships (e.g., ‘blicket detectors’; Gopnik et al., 2001) and the identification of complex causes, even in the case of illness, may depend in part on domain-general neural mechanisms”).

d) Because the magic trials are excluded from our analyses, it is unclear how the imbalance in the number of magic trials could influence the results and our interpretation of them. We note that the number of catch trials in standard target detection paradigms are sometimes much lower than the number of target trials in each condition (e.g., Pallier et al., 2011).

(4) Another imbalance is present in the design of this study: the number of trials per category is not the same in each run of the main task. This imbalance does not seem to be accounted for in the 1st-level GLM and renders a bit problematic the subsequent use of MVPA.

Each condition is shown either 6 or 7 times per run (maximum difference of 1 trial between conditions), and the number of trials per condition is equal across the whole experiment: each condition is shown 7 times in two of the runs and 6 times four of the runs. This minor design imbalance is typical of fMRI experiments and should not impact our interpretations of the data, particularly because we average responses from each condition within a run before submitting them to MVPA.

(5) The main claim of the authors, encapsulated by the title of the present manuscript, is not tested directly. While the authors included in their protocol independent localizers for mentalizing, language, and logic, they did not include an independent localizer for "animacy". As such, they cannot provide a within-subject evaluation of their claim, which is entirely based on the presence of a partial overlap in PC (which is also involved in a wide range of tasks) with previous results on animacy.

We respectfully disagree with this assertion. Our primary analysis uses a within-subject leave-one-run-out approach. This approach allows us to use part of the data itself to localize animacy-relevant causal responses in the PC without engaging in ‘double-dipping’ or statistical non-independence (Vul & Kanwisher, 2011). We also use the mentalizing network localizer as a partial localizer for animacy. This is because the control condition (physical reasoning) does not include references to people or any animate agents (Supplementary Figures 1 and 15). We now clarify this point in Methods section of the paper (see below).

From the Methods: “To test the relationship between neural responses to inferences about the body and the mind, and to localize animacy regions, we used a localizer task to identify the mentalizing network in each participant (Saxe & Kanwisher, 2003; Dodell-Feder et al., 2011; http://saxelab.mit.edu/use-our-efficient-false-belief-localizer)...Our physical stories incorporated more vivid descriptions of physical interactions and did not make any references to human agents, enabling us to use the mentalizing localizer as a localizer for animacy.”

**Reviewer #3 (Public review):**
Summary:This study employed an implicit task, showing vignettes to participants while a bold signal was acquired. The aim was to capture automatic causal inferences that emerge during language processing and comprehension. In particular, the authors compared causal inferences about illness with two control conditions, causal inferences about mechanical failures and non-causal phrases related to illnesses. All phrases that were employed described contexts with people, to avoid animacy/inanimate confound in the results. The authors had a specific hypothesis concerning the role of the precuneus (PC) in being sensitive to causal inferences about illnesses.These findings indicate that implicit causal inferences are facilitated by semantic networks specialized for encoding causal knowledge.Strengths:The major strength of the study is the clever design of the stimuli (which are nicely matched for a number of features) which can tease apart the role of the type of causal inference (illness-causal or mechanical-causal) and the use of two localizers (logic/language and mentalizing) to investigate the hypothesis that the language and/or logical reasoning networks preferentially respond to causal inference regardless of the content domain being tested (illnesses or mechanical).Weaknesses:I have identified the following main weaknesses:(1) Precuneus (PC) and Temporo-Parietal junction (TPJ) show very similar patterns of results, and the manuscript is mostly focused on PC (also the abstract). To what extent does the fact that PC and TPJ show similar trends affect the inferences we can derive from the results of the paper? I wonder whether additional analyses (connectivity?) would help provide information about this network.

We thank the reviewer for this suggestion. While the PC shows the most robust univariate preference for illness inferences compared to both mechanical inferences and noncausal vignettes, the TPJ also shows a preference for illness inferences compared to mechanical inferences in individual-subject fROI analysis. However, as we mention in the Results section, the TPJ does not show a preference for illness inferences compared to noncausal vignettes, suggesting that the TPJ is selective for animacy but may not be as sensitive to causal knowledge about animacy-specific processes. When describing our results, we refer to the ‘animacy network’ (i.e., PC and TPJ) but also highlight that the PC exhibited the most robust responses to illness inferences (from the Results: “Inferring illness causes preferentially recruited the animacy semantic network, particularly the PC”; from the Discussion: “We find that a semantic network previously implicated in thinking about animates, particularly the precuneus (PC), is preferentially engaged when people infer causes of illness…”). We did not collect resting state data that would enable a connectivity analysis, as the reviewer suggests. This is an interesting direction for future work.

(2) Results are mainly supported by an univariate ROI approach, and the MVPA ROI approach is performed on a subregion of one of the ROI regions (left precuneus). Results could then have a limited impact on our understanding of brain functioning.

The original and current versions of the paper include results from multiple multivariate analyses, including whole-cortex searchlight MVPA and individual-subject fROI MVPA performed in multiple search spaces (see Supplementary Figures 10 and 11, Supplementary Tables 2 and 3).

We note that our preregistered predictions focused primarily on univariate differences. This is because the current study investigates neural responses to inferences, and univariate increases in activity is thought to reflect the processing of such inferences. We use multivariate analyses to complement our primary univariate analyses. However, given that we observe significant univariate effects and that multivariate analyses are heavily influenced by significant univariate effects (Coutanche, 2013; Kragel et al., 2012; Hebart & Baker, 2018; Woolgar et al., 2014; Davis et al., 2014; Pakravan et al., 2022), our univariate results constitute the main findings of the paper.

(3) In all figures: there are no measures of dispersion of the data across participants. The reader can only see aggregated (mean) data. E.g., percentage signal changes (PSC) do not report measures of dispersion of the data, nor do we have bold maps showing the overlap of the response across participants. Only in Figure 2, we see the data of 6 selected participants out of 20.

We thank the reviewer for this suggestion. We now include graphs depicting the dispersion of the data across participants in the following figures: Figures 1, 3, and 4, and Supplementary Figures 8, 12, and 14. We have also created 2 figures that display the overlap of univariate responses across participants (Supplementary Figures 6 and 7). These figures show that there is high overlap across participants in PC responses to illness inferences but not mechanical inferences. In addition, all participants’ results from the analysis depicted in Figure 2 are included in Supplementary Figure 3.

(4) Sometimes acronyms are defined in the text after they appear for the first time.

We thank the reviewer for pointing this out. We now define all acronyms before using them.

**Recommendations for the authors:**

**Reviewer #1 (Recommendations for the authors):**
(1) I was unable to access the pre-registration on OSF because special permission is required.

We apologize for this technical error. The preregistration is now publicly available: https://osf.io/6pnqg.

(2) The length of the MRI session is quite long (around 2 hours). It is generally discouraged to have such extended data acquisition periods, as this can affect the stability and cleanliness of the data. Did you observe any effects of fatigue or attention decline in your data?

The session was 2 hours long including 1-2 10-minute breaks. Without breaks, the scan would be approximately 1.5 hours. This is a standard length for MRI experiments. The main experiment (causal inference task) was always conducted first and lasted approximately 1 hour. Accuracy did not decrease across the 6 runs of this experiment (repeated measures ANOVA, *F(5,114)* = 1.35, *p* = .25).

(3) The last sentence of the results states: "Although MVPA searchlight analysis identified several areas where patterns of activity distinguished between causal and non-causal vignettes, all of these regions showed a preference for non-causal vignettes in univariate analysis (Supplementary Figure 5)." This statement is not entirely accurate. As I previously pointed out, the MVPA searchlight analysis is not very informative and is difficult to interpret. However, as previously suggested, there are additional steps that could be taken to better understand and interpret these results. It is incorrect to conclude that because the brain regions identified in the MVPA analyses show a preference for non-causal vignettes in univariate analyses, the multivariate results lack value. While univariate analyses may show a preference for a specific condition, multivariate analyses can reveal more fine-grained representations of multiple conditions. For a notable example, consider the fusiform face area (FFA) that shows a clear preference for faces at the univariate level but can significantly decode other categories at the multivariate level, even when faces are not included in the analysis.

The decoding analysis that the reviewer is suggesting for the current study would be analogous to identifying univariate differences between faces and places in the FFA and then decoding between faces and places and claiming that the FFA represents places because the decoding is significant. The decoding analyses enabled by our design are not equivalent to decoding within a condition (e.g., among face identities, among types of illness inferences), as the reviewer suggests above. It is not that such multivariate analyses “lack value” but that they recapitulate established univariate differences. Multivariate analyses are useful for revealing more fine-grained representations when (i) significant univariate differences are not observed, or (ii) when it is possible to decode among categories within a condition (e.g., among face identities, among types of illness inferences). We are currently collecting data that will enable us to perform within-condition decoding analyses in future work, but the design of the current study does not allow for such a comparison.

We note that the original quotation from the manuscript has been removed because it is no longer accurate. When including participant response time as a covariate of no interest in the GLM, no regions are shared across the 4 searchlight analyses comparing causal and noncausal conditions, suggesting that there are no shared neural responses to causal inference in our dataset.

**Reviewer #2 (Recommendations for the authors):**
(1) Moderating the strength of some claims made to justify the main hypothesis (e.g., "people but not machines transmit diseases to each other through physical contact").

We changed this wording so that it now reads: “Illness affects living things (e.g., people and animals) rather than inanimate objects (e.g., rocks, machines, houses).” (Introduction)

(2) Expanding the paragraph introducing the sub-question about inferring people's "body states" vs "mental states". In addition, given the order in which the hypotheses are introduced, and the results are presented, I would suggest switching the order of presentation of both localizers in the methods section and adding a quick reminder of the hypotheses that justify using these localizers.

We thank the reviewer for these suggestions. In accordance their suggestions, we have expanded the paragraph Introduction that introduces the “body states” vs. “mental states” question (see below). We have also switched the order of the localizer descriptions in the Methods section and added a sentence at the start of each section describing the relevant hypotheses (see below).

From the Introduction: “We also compared neural responses to causal inferences about the body (i.e., illness) and inferences about the mind (i.e., mental states). Both types of inferences are about animate entities, and some developmental work suggests that children use the same set of causal principles to think about bodies and minds (Carey, 1985, 1988). Other evidence suggests that by early childhood, young children have distinct causal knowledge about the body and the mind (Springer & Keil, 1991; Callanan & Oakes, 1992; Wellman & Gelman, 1992; Inagaki & Hatano, 1993; 2004; Keil, 1994; Hickling & Wellman, 2001; Medin et al., 2010). For instance, preschoolers are more likely to view illness as a consequence of biological causes, such as contagion, rather than psychological causes, such as malicious intent (Springer & Ruckel, 1992; Raman & Winer, 2004; see also Legare & Gelman, 2008). The neural relationship between inferences about bodies and minds has not been fully described. The ‘mentalizing network’, including the PC, is engaged when people reason about agents’ beliefs (Saxe & Kanwisher, 2003; Saxe et al., 2006; Saxe & Powell, 2006; Dodell-Feder et al., 2011; Dufour et al., 2013). We localized this network in individual participants and measured its neuroanatomical relationship to the network activated by illness inferences.”

From the Methods, localizer descriptions: “To test the relationship between neural responses to inferences about the body and the mind, and to localize animacy regions, we used a localizer task to identify the mentalizing network in each participant… To test for the presence of domain-general responses to causal inference in the language and logic networks (e.g., Kuperberg et al., 2006; Operskalski & Barbey, 2017), we used an additional localizer task to identify both networks in each participant.”

(3) Adding a quick analysis of lateralization to support the corresponding claim of left lateralization of responses to causal inferences.

In accordance with the reviewer’s suggestion, we now include hemisphere as a factor in all ANOVAs comparing univariate responses across conditions.

From the Results: “In individual-subject fROI analysis (leave-one-run-out), we similarly found that inferring illness causes activated the PC more than inferring causes of mechanical breakdown (repeated measures ANOVA, condition (*Illness-Causal*, *Mechanical-Causal*) x hemisphere (left, right): main effect of condition, *F(1,19)* = 19.18, *p* < .001, main effect of hemisphere, *F(1,19)* = 0.3, *p* = .59, condition x hemisphere interaction, *F(1,19)* = 27.48, *p* < .001; Figure 1A). This effect was larger in the left than in the right PC (paired samples t-tests; left PC: *t(19)* = 5.36, *p* < .001, right PC: *t(19)* = 2.27, *p* = .04)…In contrast to the animacy-responsive PC, the anterior PPA showed the opposite pattern, responding more to mechanical inferences than illness inferences (leave-one-run-out individual-subject fROI analysis; repeated measures ANOVA, condition (*Mechanical-Causal*, *Illness-Causal*) x hemisphere (left, right): main effect of condition, *F(1,19)* = 17.93, *p* < .001, main effect of hemisphere, *F(1,19)* = 1.33, *p* = .26, condition x hemisphere interaction, *F(1,19)* = 7.8, *p* = .01; Figure 4A). This effect was significant only in the left anterior PPA (paired samples t-tests; left anterior PPA: *t(19)* = 4, *p* < .001, right anterior PPA: *t(19)* = 1.88, *p* = .08).”

(4) Making public and accessible the pre-registration OSF link.

We apologize for this technical error. The preregistration is now publicly available: https://osf.io/6pnqg.

**Reviewer #3 (Recommendations for the authors):**
In all figures: there are no measures of dispersion of the data across participants. The reader can only see aggregated (mean) data. E.g., percentage signal changes (PSC) do not report measures of dispersion of the data, nor do we have bold maps showing the overlap of the response across participants. Only in Figure 2, we see the data of 6 selected participants out of 20.

We thank the reviewer for this suggestion. We now include graphs depicting the dispersion of the data across participants in the following figures: Figures 1, 3, and 4, and Supplementary Figures 8, 12, and 14. We have also created 2 figures that display the overlap of univariate responses across participants (Supplementary Figures 6 and 7). In addition, all participants’ results from the analysis depicted in Figure 2 are included in Supplementary Figure 3.

Minor(1) Figure 2: Spatial dissociation between responses to illness inferences and mental state inferences in the precuneus (PC). If the analysis is the result of the MVPA, the figure should report the fact that only the left precuneus was analyzed.

Figure 2 depicts the spatial dissociation in univariate responses to illness inferences and mental state inferences. We now clarify this in the figure legend.

(2) VOTC and PSC acronyms are defined in the text after they appear for the first time. TPJ is never defined.

We thank the reviewer for pointing this out. We now define all acronyms before using them.